# Certified Robustness for Deep Equilibrium Models via Serialized Random Smoothing

**Weizhi Gao**[1]
wgao23@ncsu.edu

**Zhichao Hou**[1]
zhou4@ncsu.edu

**Han Xu**[2]
xuhan2@arizona.edu

**Xiaorui Liu**[1*]
xliu96@ncsu.edu

[1]North Carolina State University, [2]The University of Arizona
*corresponding author

## Abstract

Implicit models such as Deep Equilibrium Models (DEQs) have emerged as promising alternative approaches for building deep neural networks. Their certified robustness has gained increasing research attention due to security concerns. Existing certified defenses for DEQs employing deterministic certification methods such as interval bound propagation and Lipschitz-bounds can not certify on large-scale datasets. Besides, they are also restricted to specific forms of DEQs. In this paper, we provide the first randomized smoothing certified defense for DEQs to solve these limitations. Our study reveals that simply applying randomized smoothing to certify DEQs provides certified robustness generalized to large-scale datasets but incurs extremely expensive computation costs. To reduce computational redundancy, we propose a novel Serialized Randomized Smoothing (SRS) approach that leverages historical information. Additionally, we derive a new certified radius estimation for SRS to theoretically ensure the correctness of our algorithm. Extensive experiments and ablation studies on image recognition demonstrate that our algorithm can significantly accelerate the certification of DEQs by up to 7x almost without sacrificing the certified accuracy. Our code is available at https://github.com/WeizhiGao/Serialized-Randomized-Smoothing.

## 1 Introduction

The recent development of implicit layers provides an alternative and promising perspective for neural network design (Amos & Kolter, 2017; Chen et al., 2018; Agrawal et al., 2019; Bai et al., 2019, 2020; El Ghaoui et al., 2021). Different from traditional deep neural networks (DNNs) that build standard explicit deep learning layers, implicit layers define the output as the solution to certain closed-form functions of the input. These implicit layers can represent infinitely deep neural networks using only one single layer that is defined implicitly. The unique definition endows implicit models with the capability to model continuous physical systems, a task that traditional DNNs cannot accomplish (Chen et al., 2018). Additionally, the implicit function theorem enhances memory efficiency by eliminating the need to store intermediate states during forward propagation (Chen et al., 2018; Bai et al., 2019). Furthermore, implicit models offer a valuable accuracy-efficiency trade-off, making them adaptable to varying application requirements (Chen et al., 2018). These advantages underscore the significant research value of implicit models.

Deep equilibrium models (DEQs) are one promising class of implicit models that construct the output as the solution to input-dependent fixed-point problems (Bai et al., 2019). With a fixed-point solver, DEQs can be seen as infinite-depth and weight-tied neural networks. The modern DEQ-based architectures have shown comparable or even surpassing performance compared with traditional explicit models (Bai et al., 2019, 2020; Gu et al., 2020; Chen et al., 2022). Due to the superior

38th Conference on Neural Information Processing Systems (NeurIPS 2024).

performance of DEQs, their adversarial robustness has gained increasing research interest. Recent research has revealed that DEQs also suffer from similar vulnerabilities as traditional DNNs, which raises security concerns (Gurumurthy et al., 2021; Yang et al., 2022). Multiple works propose empirical defenses to improve the adversarial robustness of DEQs using regularization methods (Chu et al., 2023; El Ghaoui et al., 2021) and adversarial training (Yang et al., 2023; Gurumurthy et al., 2021; Yang et al., 2022). These empirical defenses measure models' adversarial robustness by the robust performance against adversarial attacks. However, they do not provide rigorous security guarantees and often suffer from the risk of a false sense of security (Athalye et al., 2018), leading to tremendous challenges for reliable robustness evaluation.

As alternatives to empirical defenses, certified defenses aim to provide theoretical robustness guarantees. It is worth noting that certified defenses certify that no adversarial example can ever exist within a neighborhood of the test sample regardless of the attacks, providing reliable robustness measurements and avoiding the false sense of security caused by weak attacking algorithms. Recent works have explored interval bound propagation (IBP) (Wei & Kolter, 2021; Li et al., 2022) and Lipschitz bounding (LBEN) (Havens et al., 2023; Jafarpour et al., 2021) for certifiable DEQs. However, IBP usually estimates a loose certified radius due to the error accumulation in deep networks (Zhang et al., 2021) and the global Lipschitz constant tends to provide a conservative certified radius (Huang et al., 2021). Due to the conservative certification, IBP and LBEN generate trivial certified radii (namely, close to 0) in some cases such as deep networks, especially in large-scale datasets (e.g., ImageNet) (Zhang et al., 2021; Li et al., 2023). Moreover, the design of IBP and LBEN relies on specific forms of DEQs and can not be customized to various model architectures, restricting the application of these methods.

Given the inherent limitations of existing works, the objective of this paper is to explore the certified robustness of DEQs via randomized smoothing for the first time. Randomized smoothing approaches construct smoothed classifiers from arbitrary base classifiers and provide certified robustness via statistical arguments (Cohen et al., 2019) based on the Monte Carlo probability estimation. Therefore, compared with IBP and LBEN methods, randomized smoothing has better flexibility in certifying the robustness of various DEQs of different architectures. More importantly, the probabilistic certification radius provided by randomized smoothing can be larger and generalized to large-scale datasets.

Our study reveals that applying randomized smoothing to certify DEQs can indeed provide better certified accuracy but it incurs significant computation costs due to the expensive *fixed point solvers* in DEQs and *Monte Carlo estimation* in randomized smoothing. For instance, certifying the robustness of one $256 \times 256$ image in ImageNet dataset with a typical DEQ takes up to 88.33 seconds. This raises significant efficiency challenges for the application of certifiable DEQs in real-world applications. In this paper, we further delve into the computational efficiency of randomized smoothing certification of DEQs. Our analysis reveals the computation redundancy therein, and we propose an effective approach, named Serialized Random Smoothing (SRS). Importantly, the certified radius and theoretical guarantees of vanilla randomized smoothing can not be applied in SRS. Therefore, we develop a new certified radius with theoretical guarantees for the proposed SRS. Our method tremendously accelerates the certification of DEQs by leveraging their unique property and reducing the computation redundancy of randomized smoothing. The considerable acceleration of SRS-DEQ allows us to certify DEQs on large-scale datasets such as ImageNet, which is not possible in previous works. In a nutshell, our contributions are as follows:

- We provide the first exploration of randomized smoothing for certifiable DEQs. Our study reveals significant computation challenges in such certification, and we provide insightful computation redundancy analyses.
- We propose a novel Serialized Randomized Smoothing approach to significantly accelerate the randomized smoothing certification for DEQs and corresponding certified radius estimation with new theoretical guarantees.
- We conduct extensive experiments on CIFAR-10 and ImageNet to show the effectiveness of our SRS-DEQ. Our experiments indicate that SRS-DEQ can speed up the certification of DEQs up to $7\times$ almost without sacrificing the certified accuracy.

## 2 Background

In this section, we provide necessary technical background for DEQs and Randomized Smoothing.

## 2.1 Deep Equilibrium Models

**Implicit formulation.** Traditional feedforward neural networks usually construct forward feature transformations using fixed-size computation graphs and explicit functions $\mathbf{z}^{l+1} = f_{\theta_l}(\mathbf{z}^l)$, where $\mathbf{z}^l$ and $\mathbf{z}^{l+1}$ are the input and output of layer $f_{\theta_l}(\cdot)$ with parameter $\theta_l$. DEQs, as an emerging class of implicit neural networks, define their output as the fixed point solutions of nonlinear equations:

$$\mathbf{z}^* = f_\theta(\mathbf{z}^*, \mathbf{x}), \tag{1}$$

where $\mathbf{z}^*$ is the output representation of implicit neural networks and $\mathbf{x}$ is the input data. Therefore, the computation of DEQs for each input data point $\mathbf{x}$ requires solving a fixed-point problem to obtain the representation $\mathbf{z}^*$.

**Fixed-point solvers.** Multiple fixed-point solvers have been adopted for DEQs, including the naive solver, Anderson solver, and Broyden solver (Geng & Kolter, 2023). The naive solver directly repeats the fixed-point iteration until it converges:

$$\mathbf{z}^{l+1} = f(\mathbf{z}^l, \mathbf{x}). \tag{2}$$

More details about these solvers can be referred to in the work (Cohen et al., 2019). In this paper, we denote all solvers as follows:

$$\mathbf{z} = \text{Solver}(f, \mathbf{x}, \mathbf{z}^0), \tag{3}$$

where $\mathbf{z}^0$ is the initial feature state that is taken as $\mathbf{0}$ in DEQs. All the solvers end the iteration if the estimation error $f(\mathbf{z}) - \mathbf{z}$ of the fixed point reaches a given tolerance error or a maximum iteration threshold $L$.

## 2.2 Randomized Smoothing

Randomized smoothing (Cohen et al., 2019) is a certified defense technique that guarantees $\ell_2$-norm certified robustness. Given an arbitrary base classifier $f(\cdot)$, we construct a smoothed classifier $g(\cdot)$:

$$g(\mathbf{x}) = \arg\max_{c \in \mathcal{Y}} \mathbb{P}(f(\mathbf{x} + \epsilon) = c), \tag{4}$$

$$\epsilon \sim \mathcal{N}(\mathbf{0}, \sigma^2 \mathbf{I}), \tag{5}$$

where $\mathcal{Y}$ is the label space, and $\sigma^2$ is the variance of Gaussian distribution. Intuitively, the smoothed classifier outputs the most probable class over a Gaussian distribution. If we denote $p_A$ and $p_B$ as the probabilities of the most probable class $c_A(x)$ and second probable class $c_B(x)$, Neyman-Pearson theorem (Neyman & Pearson, 1933) provides a $\ell_2$-norm certified radius $R$ for the smoothed classifier $g$:

$$g(\mathbf{x} + \delta) = c_A(x) \text{ for all } \|\delta\|_2 < R, \tag{6}$$

$$\text{where } R = \frac{\sigma}{2}(\Phi^{-1}(\underline{p_A}) - \Phi^{-1}(\overline{p_B})). \tag{7}$$

Here $\Phi(\mathbf{x})$ is the inverse of the standard Gaussian cumulative distribution function, and $\underline{p_A}, \overline{p_B} \in [0, 1]$ satisfy:

$$\mathbb{P}(f(\mathbf{x} + \epsilon) = c_A(x)) \geq \underline{p_A} \geq \overline{p_B} \geq \max_{c \neq c_A(x)} \mathbb{P}(f(\mathbf{x} + \epsilon) = c).$$

In practice, $\underline{p_A}$ and $\overline{p_B}$ are estimated using the Monte Carlo method. It is crucial to maintain the independence of each prediction in the simulation to ensure the correctness of randomized smoothing.

# 3 Serialized Randomized Smoothing

In this section, we begin by revealing the computation challenges associated with certifying DEQs using Randomized Smoothing. Subsequently, we propose Serialized Random Smoothing (SRS), a novel approach to remarkably enhance the efficiency of DEQ certification. However, directly applying the estimated radius of the standard randomized smoothing to SRS breaks the theoretical guarantee of certified robustness. To address this challenge, we develop the correlation-eliminated certification technique to estimate the radius in SRS.

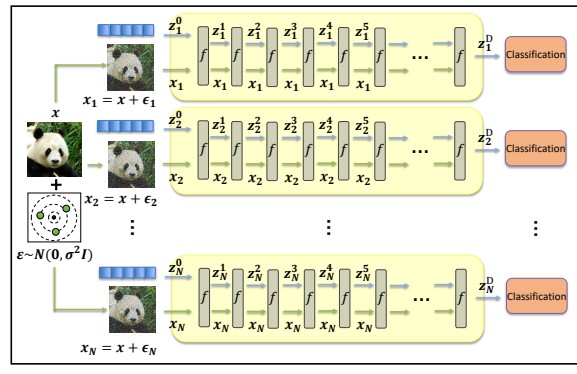 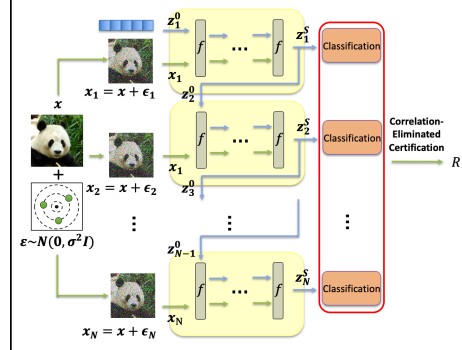

(a) Standard Randomized Smoothing          (b) Serialized Randomized Smoothing

Figure 1: Illustrations of the standard DEQ and our SRS-DEQ. The representation for each sample goes through $D$ layers in standard DEQ. Our SRS-DEQ uses the previous representation as the initialization and converges to the fixed point with a few layers ($S \ll D$). After get all the predictions, SRS-DEQ makes use of correlation-eliminated certification to estimate the certified radius.

## 3.1 Computation Challenges

According to Eq. (7), we need to estimate the lower bound probability $\underline{p_A}$ and upper bound probability $\overline{p_B}$. The appliance of Monte Carlo sampling rescues as follows:

$$\mathbb{P}(f(\mathbf{x} + \epsilon) = c) \approx \frac{1}{N} \sum_{i}^{N} \mathbf{1}\{f(\mathbf{x} + \epsilon_i) = c\}, \qquad (8)$$

where $\mathbf{1}\{\cdot\}$ is the indicator function, and $\epsilon_i \sim \mathcal{N}(\mathbf{0}, \sigma^2 \mathbf{I})$ is the $i$-th random perturbation sampled from Gaussian distribution. However, it introduces computation challenges due to the large sampling number $N$. Our empirical study (Section 4.2) indicates that applying randomized smoothing to certify MDEQ (Bai et al., 2020) on one $32 \times 32$ image in CIFAR-10 takes 12.89 seconds, and 88.33 seconds for one $256 \times 256$ image in ImageNet. The computation challenges raise tremendous limitations in real-world applications.

We provide an analysis for the slow randomized smoothing certification of DEQs. First, each forward iteration of DEQs can be very expensive. This is because DEQs are typically weight-tied neural networks so one layer $f(\mathbf{z}, \mathbf{x})$ of DEQs needs to be complex to maintain the expressiveness. Second, the fixed-point solver needs many iterations for convergence. To maintain the best performance, the solvers usually set a small value for the error tolerance (e.g., 0.001). Although second-order solvers like Broyden's method have faster convergence, their computation cost per iteration is higher. Third, the Monte Carlo estimation in randomized smoothing further exacerbates the expensive inference of DEQs, leading to significant computation challenges, as shown in Figure 1a.

## 3.2 Serialized Randomized Smoothing

As introduced in Section 3.1, the Monte Carlo method is employed in randomized smoothing to estimate the prediction probability, typically necessitating over $10,000$ times inference computation for certifying one data point. Despite the independent sampling of Gaussian noises, these noises $\{\epsilon_i\}$ are added to the same certified data point $\mathbf{x}$ to form noisy data samples $\{\mathbf{x} + \epsilon_i\}$. Notably, these samples are numerically and visually similar to each other as can be seen in Figure 1. Moreover, in randomized smoothing, the base classifier is trained on Gaussian-augmented data to be resistant to noises added to data points, yielding robust features and base classifiers. Therefore, the feature representation of these noisy samples computed in the forward computation of DEQs shares significant similarities, resulting in a substantial computation redundancy in the fixed-point solvers. The computation redundancy contributes to the inefficient DEQ certification with randomized smoothing as a primary factor. Consider a DEQ with 50 layers as an illustrative example. In the Monte Carlo estimation with $N = 10,000$, it requires the forward computation of $50 \times 10,000 = 500,000$ layers. However, if we can estimate the intermediate representation at the $45^{th}$ layer, the required forward iterations reduce to $5 \times 10,000 = 50,000$ layers, bringing a $10\times$ acceleration.

Motivated by the above challenges and analyses, we propose a novel solution, *Serialized Randomized Smoothing (SRS)*, to effectively reduce the computation redundancy in the certification of DEQs. The key idea of Serialized Randomized Smoothing is to accelerate the convergence of fixed-point solvers of DEQs by harnessing historical feature representation information $\mathbf{z}$ computed from different noisy samples, thereby mitigating redundant calculations. While the hidden representation $\mathbf{z}^0$ is initialized as $\mathbf{0}$ in standard DEQs (Bai et al., 2021a), we propose to choose a better initialization of $\mathbf{z}^0$ to accelerate the convergence of DEQs (Bai et al., 2021a), which can potentially reduce the number of fixed-point iteration $S$ and computation cost. Specifically, our Serialized Randomized Smoothing approach leverages the representation $\mathbf{z}_i^S$ computed from the previous noisy sample $\mathbf{x} + \epsilon_i$ as the initial state of the fixed-point solver of the next noisy sample:

$$\mathbf{z}_i^S = \text{Solver}(f, \mathbf{x} + \epsilon_i, \mathbf{z}_{i-1}^S), \tag{9}$$

where $i = 1, 2, \ldots, N$. As shown in Figure 1b, due to the similarity between $\mathbf{z}_{i-1}^*(\approx \mathbf{z}_{i-1}^S)$ and $\mathbf{z}_i^*(\approx \mathbf{z}_i^S)$ as analyzed in the motivation, it only takes a few fixed-point iterations to adjust the feature representation from $\mathbf{z}_{i-1}^S$ to $\mathbf{z}_i^*(\approx \mathbf{z}_i^S)$, which significantly accelerates the prediction of DEQs.

Though a better initialization accelerates the inference of DEQs, it introduces an unnecessary correlation within the framework of randomized smoothing. In standard randomized smoothing, each prediction is made independently. However, the predictions are linked through previous fixed points as defined by $\text{Solver}(f, \mathbf{x} + \epsilon_i, \mathbf{z}_{i-1}^S)$. To exemplify this, consider an extreme case where the solver functions as an identity mapping. In such a case, all subsequent predictions merely replicate the first prediction. This pronounced correlation effectively reduces the process to an amplification of the first prediction, breaking the confidence estimation for Monte Carlo. Therefore, we develop a new estimation of the certified radius with theoretical guarantees in the next subsection.

### 3.3 Correlation-Eliminated Certification

The primary challenge is to confirm how much the initialization of the fixed-point solver influences the final predictions. For different data samples $\mathbf{x} + \epsilon_i$ and initialization $\mathbf{z}_i^S$, the cases can be different depending on the complex loss landscape of the fixed-point problem and the strength of the solver. Nonetheless, comparing all predictions from SRS with standard predictions, which necessitate numerous inference steps, is impractical. Such a comparison contradicts the fundamental requirement for efficiency in this process.

To maintain the theoretical guarantee of randomized smoothing, we propose correlation-elimination certification to obtain a conservative estimate of the certified radius. The core idea involves discarding those samples that are misclassified as the most probable class, $c_A(x)$, during the Monte Carlo process. Let $p_m$ represent the probability that a sample is predicted as class $c_A(x)$ using SRS but falls into other classes with the standard DEQ. We can drop the misclassified samples as follows:

$$N_A^E = N_A - p_m N_A, \tag{10}$$

where $N_A$ represents the count of samples predicted as class $c_A(x)$ and $N_A^E$ refers to the subset of these effective samples that are predicted as class $c_A(x)$. Utilizing $N_A^E$ and $N$, we are ultimately able to estimate the certified radius. For the reason that $p_m$ is intractable, we employ an additional hypothesis test using a limited number of samples to approximate its upper bound. During the Monte Carlo sampling of SRS, we randomly select $K$ of samples (a small number compared to $N$) along with their corresponding predictions, which are then stored as $\mathbf{X}_m$ and $\mathbf{Y}_m$, respectively. After the Monte Carlo sampling, these samples, $\mathbf{X}_m$, are subjected to predictions using the standard DEQ to yield the labels $\mathbf{Y}_g$, which serve as the ground truth. Mathematically, we estimate $\overline{p_m}$ as follows:

$$N_1 = \sum_{i=1}^{K} \mathbf{1}\{Y_m = Y_g \text{ and } Y_g = c_A(x)\}, \tag{11}$$

$$N_2 = \sum_{i=1}^{K} \mathbf{1}\{Y_m = c_A(x)\}, \tag{12}$$

$$\overline{p_m} = 1 - \text{LowerConfBound}(N_1, N_2, 1 - \tilde{\alpha}), \tag{13}$$

where $\tilde{\alpha} = \alpha/2$ is for keeping the confidence level of the two-stage hypothesis test. Besides, $\text{LowerConfBound}(k, n, 1 - \alpha)$ returns a one-sided $(1 - \alpha)$ lower confidence interval for the Binomial parameter $p$ given that $k \sim \text{Binomial}(n, p)$. In other words, it returns some number $\underline{p}$ for which $\underline{p} \leq p$ with probability at least $1 - \alpha$ over the sampling of $k \sim \text{Binomial}(n, p)$. Intuitively, a smaller

$\overline{p_m}$ indicates a higher consistency between the predictions of SRS and those of the standard DEQ, yielding a greater number of effective samples. To enhance comprehension, we include an example in Appendix I to demonstrate the workflow of correlation-eliminated certification. In the end, we estimate the certified radius with the following equation:

$$\underline{p_A} = \text{LowerConfBound}(N_A^E, N, 1 - \tilde{\alpha}) \tag{14}$$

$$R = \sigma \Phi^{-1}(\underline{p_A}) \tag{15}$$

To implement SRS-DEQ efficiently, we stack the noisy samples into mini-batches for faster parallel computing as shown in Algorithm 1. Given a certified point, we sample batch-wise noisy data. After solving the fixed-point problem for the first batch, the subsequent fixed-point problem is initialized with the solution of the previous one. By counting the effective predictions using Eq. (10), the algorithm finally returns the certified radius as in standard randomized smoothing (Cohen et al., 2019). The following Theorem 3.1 theoretically guarantees the correctness of our algorithm (proof available in Appendix A):

**Theorem 3.1** (Correlation-Eliminated Certification). *If Algorithm 1 returns a class $\hat{c}_A(x)$ with a radius R calculated by equation 14 and 15, then the smoothed classifier g predicts $\hat{c}_A(x)$ within radius R around* **x***: $g(\mathbf{x} + \delta) = g(\mathbf{x})$ for all $\|\delta\| < R$, with probability at least $1 - \alpha$.*

## 4 Experiments

In this section, we conduct comprehensive experiments in the image classification tasks to demonstrate the effectiveness of the proposed SRS-MDEQ. First, we introduce the experimental settings in detail. Then we present certification on CIFAR-10 and ImageNet datasets to demonstrate the certified accuracy and efficiency. Finally, we provide comprehensive ablation studies to understand its effectiveness.

### 4.1 Experiment Settings

**Datasets.** We use two classical datasets in image recognition, CIFAR-10 (Krizhevsky et al., 2009) and ImageNet (Russakovsky et al., 2015), to evaluate the certified robustness. It is crucial to emphasize that this is the first attempt to certify the robustness of DEQs on such a large-scale dataset.

**DEQ Architectures and Solvers.** We select MDEQ with Jacobian regularization (Bai et al., 2020), a type of DEQs specially designed for image recognition, to serve as the base classifier in randomized smoothing. Specifically, we choose MDEQ-SMALL and MDEQ-LARGE for CIFAR-10, and MDEQ-SMALL for ImageNet. To obtain a satisfactory level of certified accuracy, all the base classifiers are trained on the Gaussian augmented noise data with mean $0$ and variance $\sigma^2$. Detailed information regarding the model configuration and training strategy is available in Appendix B.

We closely follow the solver setting in MDEQ (Bai et al., 2020). For the standard MDEQ on CIFAR-10, we use the Anderson solver with the step of $\{1, 5, 30\}$. For the standard MDEQ on ImageNet, we use the Broyden solver with the step of $\{1, 5, 14\}$. We apply Anderson and Naive solvers on CIFAR-10 and Broyden solver on ImageNet for the proposed SRS-MDEQ with the step of $\{1, 3\}$. We adopt a warm-up technique, where we use multi-steps to solve the fixed-point problem for the first batch in Algorithm 1. The warm-up steps for our SRS-MDEQ are set as 30 and 14 steps for CIFAR-10 and ImageNet, respectively. The details of warm-up strategy are shown in Appendix K. For notation simplicity, we use a number after the algorithm name to represent the number of layers of the model, and we use "**N**", "**A**", and "**B**" to denote the Naive, Anderson, and Broyden solvers. For instance, SRS-MDEQ-3A denotes SRS-MDEQ method with 3 steps of Anderson iterations.

**Randomized smoothing.** Following the setting in randomized smoothing (Cohen et al., 2019), we use four noise levels to construct smoothed classifiers: $\{0.12, 0.25, 0.50, 1.00\}$. We report the *approximate certified accuracy* as in (Cohen et al., 2019), which is defined as the fraction of the test data that is both correctly classified and certified with a $\ell_2$-norm certified radius exceeding a radius threshold $r$. In our experiments, we set the failure rate as $\alpha = 0.001$ and the sampling number as $N = 10,000$ in the Monte Carlo method, unless specified otherwise. All the experiments are conducted on one A100 GPU.

**Baselines.** We majorly use standard Randomized Smoothing for MDEQs as our baseline for comparison. It is also important to compare our method with state-of-the-art certified defenses. Note that the

results of randomized smoothing are not entirely comparable to deterministic methods. Although randomized smoothing can provide better certified radii, its certification is probabilistic, even if the certified probability is close to 100%. Therefore, we only show the comparison in Appendix D for reference. However, our results indicate that the certified radii can be more promising in certain cases when using deterministic methods as references.

## 4.2 Certification on CIFAR-10 and ImageNet

Table 1: Certified accuracy and running time of one image for the MDEQ-LARGE on CIFAR-10. The best certified accuracy for each radius is in bold and the time is compared with MDEQ-30A.

| Model \ Radius | 0.0 | 0.25 | 0.5 | 0.75 | 1.0 | 1.25 | 1.5 | ACR | Time (s) |
|---|---|---|---|---|---|---|---|---|---|
| MDEQ-1A | 28% | 19% | 13% | 8% | 5% | 3% | 1% | 0.27 | 1.06 |
| MDEQ-5A | 50% | 41% | 32% | 21% | 15% | 10% | 6% | 0.47 | 2.59 |
| MDEQ-30A | **67%** | **55%** | 45% | 33% | 23% | 16% | 12% | 0.62 | 12.89 |
| SRS-MDEQ-1N | 61% | 52% | 44% | 31% | 22% | 15% | 11% | 0.57 | 1.02 (**13**×) |
| SRS-MDEQ-1A | 63% | 53% | **45%** | 32% | 22% | **16%** | **12%** | 0.59 | 1.79 (**7**×) |
| SRS-MDEQ-3A | 66% | 54% | 45% | **33%** | **23%** | 16% | 11% | 0.62 | 2.55 (**5**×) |

Table 2: Certified accuracy and running time of one image for the MDEQ-SMALL on CIFAR-10. The best certified accuracy for each radius is in bold and the time is compared with MDEQ-30A.

| Model \ Radius | 0.0 | 0.25 | 0.5 | 0.75 | 1.0 | 1.25 | 1.5 | ACR | Time (s) |
|---|---|---|---|---|---|---|---|---|---|
| MDEQ-1A | 21% | 17% | 13% | 10% | 6% | 4% | 1% | 0.23 | 0.28 |
| MDEQ-5A | 52% | 42% | 29% | 21% | 11% | 7% | 3% | 0.45 | 0.64 |
| MDEQ-30A | **62%** | 50% | 38% | **30%** | **22%** | **13%** | **9%** | 0.59 | 3.08 |
| SRS-MDEQ-1N | 47% | 38% | 28% | 19% | 11% | 6% | 2% | 0.41 | 0.27 (**11**×) |
| SRS-MDEQ-1A | 60% | 47% | 36% | 27% | 17% | 12% | 8% | 0.56 | 0.46 (**7**×) |
| SRS-MDEQ-3A | 60% | **50%** | **38%** | 29% | 21% | 12% | 8% | 0.59 | 1.14 (**3**×) |

Table 3: Certified accuracy and running time of one image for the MDEQ-SMALL on ImageNet. The best certified accuracy for each radius is in bold and the time is compared with MDEQ-14B.

| Model \ Radius | 0.0 | 0.5 | 1.0 | 1.5 | 2.0 | 2.5 | 3.0 | Time (s) |
|---|---|---|---|---|---|---|---|---|
| MDEQ-1B | 2% | 2% | 1% | 1% | 1% | 1% | 0% | 7.30 |
| MDEQ-5B | 39% | 33% | 28% | 23% | 19% | 15% | 11% | 31.77 |
| MDEQ-14B | **45%** | 39% | 33% | 28% | 22% | 17% | 11% | 88.33 |
| SRS-MDEQ-1B | 40% | 34% | 32% | 27% | 21% | 16% | 10% | 15.21 (**6**×) |
| SRS-MDEQ-3B | 44% | **39%** | **33%** | **28%** | **22%** | **17%** | **11%** | 27.48 (**3**×) |

We compare the certified accuracy and the running time of standard MDEQ and our SRS-MDEQ across various layers to further validate the efficiency and robustness of the models. The experimental results of the large and small architectures on the CIFAR-10 with $\sigma = 0.5$ are presented in Tables 1 and 2, and the results on Imagenet with $\sigma = 1.0$ are shown in Table 3. The results for models using different values of $\sigma$ are provided in the Appendix E . Based on these results, we make the following observations from several aspects.

**Number of layers.** We delve into a detailed study of the impact of layers in MDEQ. The results in Table 1, Table 2, and Table 3 indicate that the certified accuracy of both MDEQ and SRS-MDEQ increases with the increase of layers. Moreover, with a few layers, our SRS-MDEQ-1 and SRS-MDEQ-3 can significantly outperform the MDEQ-1 and MDEQ-5 and achieve comparable performance to MDEQ-30. For instance, for CIFAR-10, SRS-MDEQ-3A can outperform MDEQ-1/MDEQ-5 by an average of 28.5%/12.1% (large) and 24.3%/8.8% (small), respectively. The results with other noise levels are shown in Appendix E.

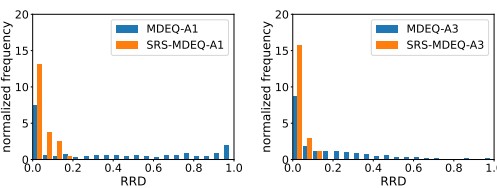

Figure 2: RRD histogram with MDEQ-LARGE with 20 bins.

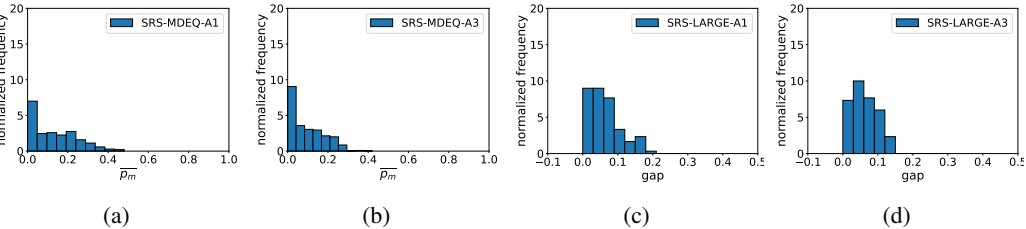

(a)                    (b)                    (c)                    (d)

Figure 3: Gap histogram of MDEQ-LARGE and $\overline{p_m}$ histogram of MDEQ-LARGE with 10 bins.

**Running Time.** The running time summarized in Table 1, Table 2, and Table 3 shows significant efficiency improvements of our SRS-DEQ method compared with standard randomized smoothing. In general, the time cost almost linearly increases with the number of layers. The standard MDEQ requires 30 layers to certify each image to a satisfactory extent, which costs 12.89 seconds per image for the large model and 3.08 seconds per image for the small model on CIFAR-10. This process leads to a heavy computational burden in the certification process. Fortunately, this process can be significantly accelerated by our SRS-MDEQ. To be specific, for CIFAR-10, large SRS-MDEQ-1N is near $11\times$ faster than large MDEQ-30A with a 2.5% certified accuracy drop. Besides, small SRS-MDEQ-3A outperforms small MDEQ-30A in efficiency by $7\times$ with only a 1% accuracy drop. On Imagenet, our SRS-MDEQ-1B can speed up the certification by $6\times$ while enhancing certified robustness compared to MDEQ-14B.

### 4.3 Ablation Study

In this section, we conduct comprehensive ablation studies to investigate the instance-level consistency and the effectiveness of our correlation-eliminated certification. We also provide more ablation studies on the hyperparameter of MDEQ solvers in Appdenix F and G. Finally, we show the empirical robustness performance of our method in Appendix L.

**Instance-level consistency.** Besides providing global measurements for the SRS-MDEQ with the certified accuracy in Section 4.2, we study how closely SRS-MDEQ matches accurate MDEQ at the instance level based on our proposed *Relative Radius Difference* (RRD). RRD compares the relative difference between the certified radius of SRS-MDEQ and the accurate MDEQ for each instance $\mathbf{x}_i$:

$$\text{RRD}(\mathbf{x}_i) = \frac{|r_b^i - r_s^i|}{r_b^i}, \tag{16}$$

where $r_b^i$ and $r_s^i$ represent the certified radius of $\mathbf{x}_i$ with MDEQ-30A and SRS-MDEQ, respectively. We compute RRD over the instances with a positive certified radius to avoid the singular value.

We present the histograms of RRD in Figure 2. As shown in Section 4.3, only with one layer, the certified radius achieved by SRS-MDEQ-1A is quite close to the accurate MDEQ since these relative differences are mostly small and close to 0, and it significantly outperforms the standard MDEQ-1A with one layer. Moreover, with 3 layers as shown in Section 4.3, the RRD values become even more concentrated around 0, which shows a very consistent certified radius with the accurate MDEQ. The instance level measurement for other settings of MDEQs are shown in Appendix H.

**Power of correlation-eliminated certification.** The correctness of our method is based on estimating the upperbound of $p_m$. In this ablation study, we investigate the effectiveness in the following two aspects. We provide additional analysis for this technique in Appendix I.

(1) The magnitude of the upperbound. A large $\overline{p_m}$ indicates that we need to drop many predictions in $\hat{c}_A$, showcasing strong correlation in SRS-DEQ. We plot the histogram of $\overline{p_m}$ to show the magnitude. Figure 3a and 3b illustrates that the majority of $\overline{p_m}$ values fall within lower intervals, even with just a single step. This trend is more pronounced with three layers, as depicted in Figure 3b. These observations suggest that an increase in the number of steps reduces the correlation in predictions, resulting in a certified radius calculated by our method closer to the one obtained through standard randomized smoothing for DEQs. The inner reason is that our approach does not necessitate the exclusion of a large number of samples for most certified points.

(2) The empirical correctness of the upperbound, i.e., if the estimated value is larger than the number of samples we should drop. For each certified point, we calculate the gap between those two values:

$$\overline{p_m} - \frac{1}{N_A} \sum_{n=1}^{N} \mathbf{1}\{y_b^n \neq y_s^n \text{ and } y_s^n = c_A\}, \tag{17}$$

where $N_A$ is the number of samples classified as $c_A$ with SRS. As shown in Figure 3c and 3d, the histogram of the gap, the value is always larger than 0, meaning that the estimation effectively covers the samples that we should drop. Moreover, the gap distribution is notably skewed towards 0. For example, more than 95% of the gaps are less than 0.2, signifying that our estimation is not only effective but also tight. More results can be found in Appendix I.

| Model\Radius | 0.0 | 0.25 | 0.5 | 0.75 | 1.0 | 1.25 | 1.5 |
|---|---|---|---|---|---|---|---|
| ResNet-110 | 65% | 54% | 41% | 32% | 23% | 15% | 9% |
| MDEQ-30A | **67%** | **55%** | 45% | 33% | 23% | 16% | 12% |
| SRS-MDEQ-3A | 66% | 54% | **45%** | **33%** | **23%** | **16%** | 11% |

Table 4: Comparison of certified accuracy for ResNet-110 and the MDEQ-LARGE architecture with $\sigma = 0.5$ on CIFAR-10. The best certified accuracy for each radius is in bold.

**Compared to explicit neural networks.** To demonstrate the superior performance of certification with DEQs, we also compare our results against those of explicit neural networks. Despite surpassing the performance of explicit neural networks is not our target, we claim the performance of DEQs can catch up with them, as shown in Table 4. We provide a comparison between DEQs and ResNet-110 under the same training and evaluation setting, and the results are consistent with those reported in (Cohen et al., 2019).

**Results on Other Randomized Smoothing methods.** In addition to the standard version of randomized smoothing (Cohen et al., 2019), there are more advanced methods available. To demonstrate the general adaptability of our approach to randomized smoothing, we conduct experiments using SmoothAdv (Salman et al., 2019). For these experiments, we utilize PGD (Kurakin et al., 2016) as the adversarial attack method, setting the number of adversarial examples during training to 4. The results, presented in Table 5, show that SmoothAdv improves certified accuracy for both standard randomized smoothing and our SRS approach.

| Model \ Radius | 0.0 | 0.25 | 0.5 | 0.75 | 1.0 | 1.25 | 1.5 |
|---|---|---|---|---|---|---|---|
| MDEQ-1A (adv) | 23% | 17% | 12% | 9% | 6% | 4% | 2% |
| MDEQ-5A (adv) | 52% | 44% | 32% | 21% | 17% | 14% | 10% |
| MDEQ-30A (adv) | **62%** | **54%** | 43% | **37%** | **30%** | **23%** | 14% |
| MDEQ-30A (standard) | 62% | 50% | 38% | 30% | 22% | 13% | 9% |
| SRS-MDEQ-1A (adv) | 60% | 43% | 35% | 27% | 18% | 14% | 9% |
| SRS-MDEQ-3A (adv) | 60% | 52% | **43%** | 36% | 29% | 22% | **14%** |

Table 5: Certified accuracy for the MDEQ-SMALL architecture with $\sigma = 0.5$ on CIFAR-10 using SmoothAdv. The maximum norm $\epsilon$ of PGD is set as 0.5 and the number of steps $T$ is set as 2.

## 5    Related Work

### 5.1    Deep Equilibrium Models

Recently, there have been many works on deep implicit models that define the output by implicit functions (Amos & Kolter, 2017; Chen et al., 2018; Bai et al., 2019; Agrawal et al., 2019; El Ghaoui et al., 2021; Bai et al., 2020; Winston & Kolter, 2020). Among these, deep equilibrium model defines the implicit layer by solving a fixed-point problem (Bai et al., 2019, 2020). There are many fundamental works investigating the existence and the convergence of the fixed point (Winston & Kolter, 2020; Revay et al., 2020; Bai et al., 2021b; Ling et al., 2023). With many advantages, DEQs achieve superior performance in many tasks, such as image recognition (Bai et al., 2020), image generation (Pokle et al., 2022), graph modeling (Gu et al., 2020; Chen et al., 2022), language modeling (Bai et al., 2019), and solving complex equations (Marwah et al., 2023). Though DEQs catch up with the performance of DNNs, the computation inefficiency borders the deployment of deep implicit models in practice (Chen et al., 2018; Dupont et al., 2019; Bai et al., 2019). Related works focus on reusing information from diffusion models and optical flows, demonstrating the effectiveness of reducing computational redundancy of DEQs (Bai & Melas-Kyriazi, 2024; Bai et al., 2022). However, this paper focuses on the certified robustness of DEQs and provides a theoretical analysis of our proposed method.

### 5.2    Certified Robustness

Empirical defenses like adversarial training are well-known in deep learning (Goodfellow et al., 2014). Some existing works improve the robustness of DEQs by applying adversarial training (Gurumurthy et al., 2021; Yang et al., 2023, 2022). Different from the empirical defense like adversarial training, certified defenses theoretically guarantee the predictions in a small ball maintain as a constant (Wong & Kolter, 2018; Raghunathan et al., 2018; Gowal et al., 2018; Cohen et al., 2019). The most common way to certify robustness is to define a convex program, which lower bounds the worst-case perturbed output of the network (Raghunathan et al., 2018; Wong & Kolter, 2018). The increasing computation complexity in high-dimension optimization hinders the generalization of these methods. Interval bound propagation (IBP) is another certification method for neural networks, which computes an upper bound of the class margin through forward propagation (Gowal et al., 2018). However, the layer-by-layer computation mode brings a potentially loose certified radius. Recently, randomized smoothing has drawn much attention due to its flexibility (Cohen et al., 2019). Randomized smoothing certifies $\ell_2$-norm robustness for arbitrary classifiers by constructing a smoothed version of the classifier. There are some existing works certifying robustness for DEQs. Most of them adapt IBP to DEQs by constructing a joint fixed-point problem (Wei & Kolter, 2021; Li et al., 2022). Others design specific forms of DEQs to control the Lipschitz constant of the models (Havens et al., 2023; Jafarpour et al., 2021). Yet, no existing work explores randomized smoothing for certifiable DEQs.

## 6    Conclusion

In this work, we provide the first exploration of randomized smoothing certification for DEQs. Our study shows that randomized smoothing for DEQs can certify more generalized architectures and be applied to large-scale datasets but it incurs significant computation costs. We delve into the computation bottleneck of this certified defense and point out the new insight of computation redundancy. We further propose a novel Serialized Random Smoothing approach to significantly reduce the computation cost by leveraging the computation redundancy. Finally, we propose a new estimation for the certified radius for our SRS. Our extensive experiments demonstrate that our algorithm significantly accelerates the randomized smoothing certification by up to $7\times$ almost without sacrificing the certified accuracy. Our discoveries and algorithm provide valuable insight and a solid step toward efficient robustness certification of DEQs. Our work significantly improves the security of artificial intelligence, especially applicable in sensitive domains, enhancing the appliance of the models and maintaining the integrity of AI-driven decisions. Though our paper speeds up the certification of DEQs with randomized smoothing, it cannot be directly applied to other architecture. We regard the speedup for the general method as our future research.

# 7 Acknowledgements

Weizhi Gao, Zhichao Hou, and Dr. Xiaorui Liu are supported by the National Science Foundation (NSF) National AI Research Resource Pilot Award, Amazon Research Award, NCSU Data Science Academy Seed Grant Award, and NCSU Faculty Research and Professional Development Award.

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

# A  Proofs of Theorem 3.1

**Theorem.** *With probability at least $1 - \alpha$ over Algorithm 1. If Algorithm 1 returns a class $\hat{c}_A$ with a radius $R$, then the smoothed classifier $g$ predicts $\hat{c}_A$ within radius $R$ around $\mathbf{x}$: $g(\mathbf{x} + \delta) = g(\mathbf{x})$ for all $\|\delta\| < R$.*

*Proof.* From the contract of the hypothesis test, we know that with the probability of at least $1 - \tilde{\alpha}$ over all the samplings $\epsilon_1, \epsilon_2, \cdots, \epsilon_N$, we have $\overline{p_m} > \mathbb{P}(y_s^i = \hat{c}_A$ and $y_b^i \neq \hat{c}_A) = p_m$, where $y_s^i$ and $y_b^i$ represent the predictions of $\mathbf{x} + \epsilon_i$ given by SRS and the standard DEQ, respectively. Denote the number of samplings as follows:

$$N_A^E = N_A - p_m N_A, \tag{18}$$

$$\hat{N}_A^E = N_A - \overline{p_m} N_A, \tag{19}$$

where $N_A^E$ is the fact number that the predictions of the standard DEQs are class $\hat{c}_A$, while $\hat{N}_A^E$ is the number we estimate. In this way, LowerConfBound($\hat{N}_A^E, N, \tilde{\alpha}$) < LowerConfBound($N_A^E, N, \tilde{\alpha}$). Suppose the standard randomized smoothing returns $\overline{R}$ with $N_A^E$ and $N$, we conclude that $R < \overline{R}$ with the probability of at least $1 - \tilde{\alpha}$. With Proposition 2 in the standard randomized smoothing (Cohen et al., 2019), $g(\mathbf{x} + \delta) = g(\mathbf{x})$ for all $\|\delta\| < \overline{R}$ for all $\|\delta\| < \overline{R}$. Denote the event that the radius of SRS is smaller than the radius of RS as $A$ and the event that the radius of RS can certify the data points $B$. We can conclude that $\mathbb{P}(\bar{A}) = \mathbb{P}(\bar{B}) = \tilde{\alpha}$ following the hypothesis tests. The final probability of successfully certifying the data point is:

$$\mathbb{P}(A \cup B) = \mathbb{P}(B) - \mathbb{P}(\bar{A} \cup B) = 1 - \mathbb{P}(\bar{B}) - \mathbb{P}(\bar{A} \cup B) \geq 1 - \mathbb{P}(\bar{B}) - \mathbb{P}(\bar{A}) = 1 - 2\tilde{\alpha} \tag{20}$$

where $\mathbb{P}(\bar{A})$ is the probability that $A$ does not happen. By setting $\tilde{\alpha} = \alpha/2$, we complete the proof. $\square$

# B  Experiment Details

## B.1  Model Architecture

Multi-resolution deep equilibrium models (MDEQ) are a new class of implicit networks that are suited to large-scale and highly hierarchical pattern recognition domains. They are inspired by the modern computer vision deep neural networks, which leverage multi-resolution techniques to learn features. These simultaneously learned multi-resolution features allow us to train a single model on a diverse set of tasks and loss functions, such as using a single MDEQ to perform both image classification and semantic segmentation. MDEQs are able to match or exceed the performance of recent competitive computer vision models, achieving high accuracy in sequence modeling.

We report the model hyperparameters in Table 6. For CIFAR-10, we utilize both MDEQ-SMALL and MDEQ-LARGE architectures, while for ImageNet, we only employ MDEQ-SMALL. Most of the hyperparameters remain consistent with the ones specified in work (Bai et al., 2021b). MDEQs define several resolutions in the implicit layer to align with ResNet in computer vision. Consequently, the primary distinction between the models lies in the channel size and resolution level. Additionally, all the models are equipped with GroupNorm, as presented in the standard MDEQ (Bai et al., 2020).

|  | CIFAR-10 | | ImageNet |
|  | SMALL | LARGE | SMALL |
| --- | --- | --- | --- |
| Input Size | $32 \times 32$ | $32 \times 32$ | $224 \times 224$ |
| Block | BASIC | BASIC | BOTTLENECK |
| Number of Branches | 3 | 4 | 4 |
| Number of Channels | [8, 16, 32] | [32, 64, 128, 256] | [32, 64, 128, 256] |
| Number of Head Channels | [7, 14, 28] | [14, 28, 56, 112] | [28, 56, 112, 224] |
| Final Channel Size | 200 | 1680 | 2048 |

Table 6: Model hyperparameters in our experiments.

## B.2 Training Setting

We report the training hyperparameters in Table 8. Following work (Bai et al., 2021b), we use Jacobian regularization to ensure stability during the training process. Moreover, to prevent overfitting, we employ data augmentation techniques such as random cropping and horizontal flipping, which are commonly utilized in various computer vision tasks.

**Gaussian Augmentation**: As mentioned in Section 3.2, randomized smoothing requires the base classifier to be robust against Gaussian noise. Therefore, we train the MDEQs with Gaussian augmentation. Following the standard randomized smoothing (Cohen et al., 2019), we augment the original data with noise sampled from $\mathcal{N}(0, \sigma^2 \mathbf{I})$, where $\sigma$ denotes the noise level in the smoothed classifier. Intuitively, the training scheme forces the base classifier to be robust to the Gaussian noise, which is used in randomized smoothing. Formally, under the cross-entropy loss, the objective is to maximize the following:

$$\sum_{i=1}^{n} \mathbb{E}_\epsilon \log \frac{\exp f_{c_i}(\mathbf{x}_i + \epsilon)}{\sum_{c \in \mathcal{Y}} \exp f_c(\mathbf{x}_i + \epsilon)}, \tag{21}$$

where $(x_i, c_i)$ is one clean data point with its ground-truth label. According to Jensen's inequality, Equation (21) is the lower bound of the following one:

$$\sum_{i=1}^{n} \log \mathbb{E}_\epsilon \frac{\exp f_{c_i}(\mathbf{x}_i + \epsilon)}{\sum_{c \in \mathcal{Y}} \exp f_c(\mathbf{x}_i + \epsilon)}. \tag{22}$$

The equation within the expectation represents the softmax output of the logits produced by the base classifier $f$. This can be seen as a soft version of the *argmax* function. Consequently, the expectation approximates the probability of class $c_i$ when Gaussian augmentation is applied. By doing so, we aim to maximize the likelihood of the smoothed classifier $g(\mathbf{x})$:

$$\sum_{i=1}^{n} \log \mathbb{P}(f(\mathbf{x}_i + \epsilon) = c_i). \tag{23}$$

To demonstrate our method is general to DEQs with any training scheme, we conduct ablation studies of Jacobian regularization. Jacobian regularization stabilizes the training of the backbones but it is not crucial for the certification (Bai et al., 2021b). The results in Table 7 show that using Jacobian regularization can help stabilize the fixed-point solvers but will almost not affect the final performance with enough fixed-point iterations. Our conclusion is consistent with Bai et al. (2021b) where the regularization does not increase the accuracy but decreases the number of fixed-point iterations. Though the experiments show that using Jacobian regularization is not crucial in the certification, we recommend to use the regularization in the training for more stable performance.

| Model \ Radius | 0.0 | 0.25 | 0.5 | 0.75 | 1.0 | 1.25 | 1.5 |
|---|---|---|---|---|---|---|---|
| MDEQ-30A (w/o Jacobian) | 63% | 51% | 38% | 29% | 19% | 13% | 7% |
| SRS-MDEQ-3A (w/o Jacobian) | 60% | 49% | 38% | 28% | 18% | 12% | 6% |
| MDEQ-30A (w Jacobian) | 62% | 50% | 38% | 30% | 22% | 13% | 9% |
| SRS-MDEQ-3A (w Jacobian) | 60% | 50% | 38% | 29% | 21% | 12% | 8% |

Table 7: Ablation study of Jacobian regularization for the MDEQ-SMALL architecture with $\sigma = 0.5$ on CIFAR-10.

## C Algorithm

In this subsection, we present the pseudo code of our algorithm in Algorithm 1. The implementation is based on mini-batch mode, which is efficient in practice.

|  | CIFAR-10 | | ImageNet |
|  | SMALL | LARGE | SMALL |
| --- | --- | --- | --- |
| Batch Size | 96 | 96 | 128 |
| Epochs | 120 | 220 | 100 |
| Optimizer | Adam | Adam | SGD |
| Learning Rate | 0.001 | 0.001 | 0.04 |
| Learning Rate Schedule | Cosine | Cosine | Cosine |
| Momentum | 0.98 | 0.98 | 0.9 |
| Weight Decay | 0.0 | 0.0 | $2 \times 10^{-5}$ |
| Jacobian Reg. Strength | 0.5 | 0.4 | 2.5 |
| Jacobian Reg. Frequency | 0.05 | 0.02 | 0.08 |

Table 8: Training hyperparameters in our experiments.

---

**Algorithm 1** Certified Radius with SRS-DEQ

---

**Require:** DEQ $f(\cdot)$, certified data $\mathbf{x}$, sampling numbers $N$, batch size $B$, failure rate $\alpha$
1: Initialize **Counts** with 0 for each class
2: Calculate hypothesis test confidence: $\hat{\alpha} = 1 - \sqrt{1 - \alpha}$
3: Fixed-point initialization: $Z^0 = \mathbf{0} \in \mathbb{R}^{B \times d}$
4: **for** $1 \leq i \leq N/B$ **do**
5:     $\mathbf{X} = [\mathbf{x}, \mathbf{x}, \ldots, \mathbf{x}]^\top \in \mathbb{R}^{B \times d}$
6:     Sample $\epsilon_j \sim \mathcal{N}(\mathbf{0}, \sigma^2 \mathbf{I})$, $\forall j = 1, 2, \cdots, B$
7:     $\tilde{\mathbf{X}} = [\mathbf{x} + \epsilon_1, \mathbf{x} + \epsilon_2, \ldots, \mathbf{x} + \epsilon_B]^\top \in \mathbb{R}^{B \times d}$
8:     $\mathbf{Z}^i = \text{Solver}(f, \tilde{\mathbf{X}}, \mathbf{Z}^{i-1})$
9:     Classify $\mathbf{Z}^i$ to get **Predictions**
10:    Store samples and labels in $\mathbf{X}_m$ and $\mathbf{Y}_m$
11:    Update **Counts** according to **Predictions**
12: **end for**
13: Predict $\mathbf{X}_m$ with the standard DEQ to get $\mathbf{Y}_g$
14: Compute the estimated $\overline{p_m}$ using Eq. (13)
15: Compute $N_A^E$ using Eq. (10)
16: Compute $R$ with **Counts** using Eq. (7)
**Return:** $R$

---

# D Comparsion with Baselines

In certified defenses, $\ell_2$-norm and $\ell_\infty$-norm are widely used. Since randomized smoothing provides $\ell_2$-norm certified radii, we choose baselines with the same certified norm. We compare the performance with the state-of-the-art DEQ certification methods on CIFAR-10 (Havens et al., 2023) including SLL (Araujo et al., 2023) and LBEN (Revay et al., 2020), which certify robustness with Lipschitz bound. To align with the numbers reported in Havens et al. (2023), we adopt the same certified radius in the table. We present the certified accuracy for the large SRS-MDEQ under different certified radii. Since the code of SLL and LBEN is not publicly available, our comparison is based on reported results in their papers for CIFAR-10.

| Model | Certification Method | $r = 0.0$ | $r = \frac{36}{255}$ | $r = \frac{72}{255}$ | $r = \frac{108}{255}$ | $r = 1.0$ |
| --- | --- | --- | --- | --- | --- | --- |
| SLL | Lipschitz Bound | 65% | 56% | 46% | 36% | 11% |
| LBEN | Lipschitz Bound | 45% | 36% | 28% | 21% | 4% |
| SRS-MDEQ-1A | Randomized Smoothing | 63% | 59% | 53% | 48% | 22% |
| SRS-MDEQ-3A | Randomized Smoothing | **66%** | **62%** | **56%** | **52%** | **25%** |

Table 9: Comparison with existing certification methods and SRS-MDEQ-LARGE. The best certified accuracy is in bold.

As shown in Table 9, our SRS-MDEQ-1 already significantly outperforms existing certification methods across all the certified radii. It also verifies that randomized smoothing defense provides tighter certification bounds. We want to emphasize again that the results of randomized smoothing are not entirely comparable to deterministic methods. Although randomized smoothing can provide better-certified radii, its certification is probabilistic, even if the certified probability is close to 100%.

# E Other Noise Levels

In the main paper, we present results using a noise level of $\sigma = 0.5$ for CIFAR-10 and $\sigma = 1.0$ for ImageNet. In this appendix, we present the results on other noise levels in Tables 10 to 17. For ImageNet, we adopt a larger noise level as suggested in paper (Cohen et al., 2019). The reason behind this choice is that images can tolerate higher levels of isotropic Gaussian noise while still preserving their high-resolution content. Since the noise level does not affect the running time, we do not show the running time in these tables.

**Tendency**: The observed performance trend aligns with standard randomized smoothing techniques. When using a smaller noise level $\sigma$, the models exhibit higher certified accuracy within a smaller radius. However, the models are unable to provide reliable certification for larger radii due to the base classifier's lack of robustness against high-level noise. For example, the smoothed classifier with $\sigma = 0.12$ on CIFAR-10 can certify up to $70\%$ accuracy within a radius of $0.25$, but its certification capability is truncated when the radius exceeds $0.5$. Conversely, when employing a larger noise level $\sigma$, the model can certify a larger radius, but this results in a drop in accuracy on clean data.

| Model \ Radius | 0.0 | 0.25 | 0.5 | 0.75 | 1.0 | 1.25 | 1.5 |
|---|---|---|---|---|---|---|---|
| MDEQ-1A | 23% | 14% | 0% | 0% | 0% | 0% | 0% |
| MDEQ-5A | 73% | 48% | 0% | 0% | 0% | 0% | 0% |
| MDEQ-30A | 86% | 68% | 0% | 0% | 0% | 0% | 0% |
| SRS-MDEQ-1N | 85% | 67% | 0% | 0% | 0% | 0% | 0% |
| SRS-MDEQ-1A | 85% | 68% | 0% | 0% | 0% | 0% | 0% |
| SRS-MDEQ-3N | 85% | 65% | 0% | 0% | 0% | 0% | 0% |
| SRS-MDEQ-3A | **86%** | **69%** | 0% | 0% | 0% | 0% | 0% |

Table 10: Certified accuracy for MDEQ-LARGE with $\sigma = 0.12$ on CIFAR-10.

| Model \ Radius | 0.0 | 0.25 | 0.5 | 0.75 | 1.0 | 1.25 | 1.5 |
|---|---|---|---|---|---|---|---|
| MDEQ-1A | 20% | 13% | 8% | 4% | 0% | 0% | 0% |
| MDEQ-5A | 33% | 21% | 15% | 10% | 0% | 0% | 0% |
| MDEQ-30A | 79% | 63% | 47% | **32%** | 0% | 0% | 0% |
| SRS-MDEQ-1N | 74% | 61% | 46% | 30% | 0% | 0% | 0% |
| SRS-MDEQ-1A | 74% | 61% | 46% | 29% | 0% | 0% | 0% |
| SRS-MDEQ-3N | 77% | 60% | **48%** | 31% | 0% | 0% | 0% |
| SRS-MDEQ-3A | **80%** | **63%** | 47% | 31% | 0% | 0% | 0% |

Table 11: Certified accuracy for MDEQ-LARGE with $\sigma = 0.25$ on CIFAR-10.

The strength of the solvers influences the certified accuracy. With a stronger Anderson solver, the SRS-MDEQ performs better than the naive one. For ImageNet, the high-resolution difficulty even requires us to use a quasi-Newton method to keep the convergence of the model. The number of steps is another crucial factor for the performance. The model has better performance no matter what solver you use.

# F MDEQs with More Steps

In the main paper, we showcase the results of our SRS limited to a maximum of 3 steps for efficiency. These results nearly match the performance of the standard randomized smoothing approach, albeit

| Model \ Radius | 0.0 | 0.25 | 0.5 | 0.75 | 1.0 | 1.25 | 1.5 |
|---|---|---|---|---|---|---|---|
| MDEQ-1A | 16% | 14% | 13% | 11% | 9% | 7% | 6% |
| MDEQ-5A | 38% | 30% | 24% | 18% | 14% | 11% | 8% |
| MDEQ-30A | **48%** | 41% | **35%** | 28% | 23% | **19%** | 15% |
| SRS-MDEQ-1N | 40% | 35% | 28% | 24% | 19% | 16% | 11% |
| SRS-MDEQ-1A | 42% | 38% | 29% | 24% | 19% | 17% | 14% |
| SRS-MDEQ-3N | 44% | 40% | 32% | 24% | 20% | 17% | 13% |
| SRS-MDEQ-3A | 46% | **41%** | 34% | **28%** | **23%** | 18% | **15%** |

Table 12: Certified accuracy for MDEQ-LARGE with $\sigma = 1.0$ on CIFAR-10.

| Model \ Radius | 0.0 | 0.25 | 0.5 | 0.75 | 1.0 | 1.25 | 1.5 |
|---|---|---|---|---|---|---|---|
| MDEQ-1A | 28% | 15% | 0% | 0% | 0% | 0% | 0% |
| MDEQ-5A | 62% | 35% | 0% | 0% | 0% | 0% | 0% |
| MDEQ-30A | **80%** | 52% | 0% | 0% | 0% | 0% | 0% |
| SRS-MDEQ-1N | 72% | 37% | 0% | 0% | 0% | 0% | 0% |
| SRS-MDEQ-1A | 75% | 44% | 0% | 0% | 0% | 0% | 0% |
| SRS-MDEQ-3N | 75% | 44% | 0% | 0% | 0% | 0% | 0% |
| SRS-MDEQ-3A | 79% | **52%** | 0% | 0% | 0% | 0% | 0% |

Table 13: Certified accuracy for MDEQ-SMALL with $\sigma = 0.12$ on CIFAR-10.

| Model \ Radius | 0.0 | 0.25 | 0.5 | 0.75 | 1.0 | 1.25 | 1.5 |
|---|---|---|---|---|---|---|---|
| MDEQ-1A | 21% | 12% | 7% | 4% | 0% | 0% | 0% |
| MDEQ-5A | 56% | 37% | 20% | 12% | 0% | 0% | 0% |
| MDEQ-30A | 72% | **54%** | **38%** | 24% | 0% | 0% | 0% |
| SRS-MDEQ-1N | 62% | 48% | 30% | 14% | 0% | 0% | 0% |
| SRS-MDEQ-1A | 69% | 52% | 35% | 19% | 0% | 0% | 0% |
| SRS-MDEQ-3N | 69% | 51% | 34% | 16% | 0% | 0% | 0% |
| SRS-MDEQ-3A | **72%** | 53% | 36% | **24%** | 0% | 0% | 0% |

Table 14: Certified accuracy for MDEQ-SMALL with $\sigma = 0.25$ on CIFAR-10.

| Model \ Radius | 0.0 | 0.25 | 0.5 | 0.75 | 1.0 | 1.25 | 1.5 |
|---|---|---|---|---|---|---|---|
| MDEQ-1A | 21% | 18% | 15% | 13% | 11% | 9% | 6% |
| MDEQ-5A | 37% | 31% | 23% | 17% | 14% | 11% | 9% |
| MDEQ-30A | 46% | **39%** | 32% | 24% | 19% | 16% | **14%** |
| SRS-MDEQ-1N | 38% | 32% | 25% | 20% | 17% | 14% | 11% |
| SRS-MDEQ-1A | 42% | 35% | 26% | 22% | 18% | 15% | 13% |
| SRS-MDEQ-3N | 44% | 35% | 27% | 21% | 17% | 15% | 12% |
| SRS-MDEQ-3A | **46%** | 38% | **32%** | **24%** | **19%** | **16%** | 13% |

Table 15: Certified accuracy for the MDEQ-SMALL with $\sigma = 1.0$ on CIFAR-10.

with slight discrepancies. This ablation study extends the process to 5 steps to evaluate potential performance improvements. We replicate the experimental settings from the main paper but with the increased step count. The outcomes, detailed in Tables 18 to 20, reveal that while there is some enhancement in performance, the gains are marginal. Given that our method prioritizes efficiency, we find that three steps are sufficient for all the experiments conducted in our paper.

| Model \ Radius | 0.0 | 0.5 | 1.0 | 1.5 | 2.0 | 2.5 | 3.0 |
|---|---|---|---|---|---|---|---|
| MDEQ-1B | 0% | 0% | 0% | 0% | 0% | 0% | 0% |
| MDEQ-5B | 54% | 40% | 0% | 0% | 0% | 0% | 0% |
| MDEQ-14B | **67%** | 52% | 0% | 0% | 0% | 0% | 0% |
| SRS-MDEQ-1B | 62% | **52%** | 0% | 0% | 0% | 0% | 0% |
| SRS-MDEQ-3B | 66% | 52% | 0% | 0% | 0% | 0% | 0% |

Table 16: Certified accuracy for the MDEQ-SMALL with $\sigma = 0.25$ on ImageNet.

| Model \ Radius | 0.0 | 0.5 | 1.0 | 1.5 | 2.0 | 2.5 | 3.0 |
|---|---|---|---|---|---|---|---|
| MDEQ-1B | 0% | 0% | 0% | 0% | 0% | 0% | 0% |
| MDEQ-5B | 47% | 38% | 29% | 22% | 0% | 0% | 0% |
| MDEQ-14B | **57%** | 46% | 37% | 27% | 0% | 0% | 0% |
| SRS-MDEQ-1B | 53% | **46%** | 37% | **27%** | 0% | 0% | 0% |
| SRS-MDEQ-3B | 55% | 46% | 37% | 27% | 0% | 0% | 0% |

Table 17: Certified accuracy for the MDEQ-SMALL with $\sigma = 0.5$ on ImageNet.

| Model \ Radius | 0.0 | 0.25 | 0.5 | 0.75 | 1.0 | 1.25 | 1.5 |
|---|---|---|---|---|---|---|---|
| MDEQ-30A | 67% | **55%** | 45% | 33% | 23% | 16% | 12% |
| SRS-MDEQ-1A | 63% | 53% | 45% | 32% | 22% | 16% | 12% |
| SRS-MDEQ-3A | 66% | 54% | 45% | 33% | 23% | 16% | 11% |
| SRS-MDEQ-5A | 66% | 54% | **45%** | **33%** | **23%** | **16%** | **12%** |

Table 18: Certified accuracy for MDEQ-SMALL with more steps on CIFAR-10 ($\sigma = 0.5$).

| Model \ Radius | 0.0 | 0.25 | 0.5 | 0.75 | 1.0 | 1.25 | 1.5 |
|---|---|---|---|---|---|---|---|
| MDEQ-30A | **62%** | 50% | 38% | 30% | 22% | **13%** | **9%** |
| SRS-MDEQ-1A | 60% | 47% | 36% | 27% | 17% | 12% | 8% |
| SRS-MDEQ-3A | 60% | 50% | 38% | 29% | 21% | 12% | 8% |
| SRS-MDEQ-5A | 61% | **50%** | **38%** | **30%** | **22%** | 12% | 8% |

Table 19: Certified accuracy for MDEQ-SMALL with more steps on CIFAR-10 ($\sigma = 0.5$).

| Model \ Radius | 0.0 | 0.25 | 0.5 | 0.75 | 1.0 | 1.25 | 1.5 |
|---|---|---|---|---|---|---|---|
| MDEQ-14B | **45%** | 39% | 33% | 28% | 22% | 17% | 11% |
| SRS-MDEQ-1B | 40% | 34% | 32% | 27% | 21% | 16% | 10% |
| SRS-MDEQ-3B | 44% | 39% | 33% | 28% | **22%** | 17% | 11% |
| SRS-MDEQ-5B | 44% | **39%** | **33%** | **28%** | 21% | **17%** | **11%** |

Table 20: Certified accuracy for MDEQ-SMALL with more steps on ImageNet ($\sigma = 1.0$).

## G  MDEQs with Different Solvers

In the main paper, our presentation of MDEQ results exclusively features the Anderson solver applied to CIFAR-10. Complementary to this, in the appendix, we provide results obtained with the naive solver on CIFAR-10 to show whether the choice of the solver significantly affects the performance. For the experiments conducted on MDEQs applied to ImageNet, our primary focus lies on the Broyden solver, as detailed in the main paper. It is worth noting that the fixed-point problem

| Model \ Radius | 0.0 | 0.25 | 0.5 | 0.75 | 1.0 | 1.25 | 1.5 |
|---|---|---|---|---|---|---|---|
| MDEQ-LARGE-30N | 64% | 53% | 42% | 31% | 21% | 13% | 11% |
| MDEQ-LARGE-30A | 67% | 55% | 45% | 33% | 23% | 16% | 12% |
| MDEQ-SMALL-30N | 60% | 47% | 36% | 29% | 19% | 11% | 8% |
| MDEQ-SMALL-30A | 62% | 50% | 38% | 30% | 22% | 13% | 9% |

Table 21: Certified accuracy of MDEQ-SMALL with different solvers on CIFAR-10.

| | Large | | Small | |
|---|---|---|---|---|
| | MDEQ | SRS-MDEQ | MDEQ | SRS-MDEQ |
| Anderson 1 | 0.6140 | 0.0266 | 0.5180 | 0.0786 |
| Anderson 3 | 0.3240 | 0.0210 | 0.2561 | 0.0324 |
| Naive 1 | 0.5219 | 0.0396 | 0.4761 | 0.2322 |
| Naive 3 | 0.1268 | 0.0425 | 0.2138 | 0.0825 |

Table 22: The mean RRD measurements over all images.

encountered in high-dimensional data introduces heightened complexities, necessitating a solver with superior convergence properties, as elaborated upon in (Bai et al., 2020).

The outcomes obtained with various solvers are detailed in Table 21. Notably, while MDEQs employing the naive solver exhibit slightly faster certification compared to those with the Anderson solver, both the large and small architectures encounter some accuracy drops, with deviations of up to $3\%$. Given that our objective is to establish MDEQs as a baseline with superior performance, we exclusively present results obtained with the Anderson solver in the main paper for meaningful comparison.

## H    Instance-Level Consistency

In the main paper, we utilize RRD as the measurement to validate the effectiveness of our method. In this appendix, we extend our evaluation to observe the performance of RRD specifically on MDEQ-SMALL architectures.

As illustrated in Figure 4, our method demonstrates strong performance for MDEQ-SMALL at the instance level. Regarding RRD, SRS-MDEQ exhibits a notable concentration of small values, indicating the efficacy of our serialized randomized smoothing in aligning radii. Further insight is gained from Figure 4b, where the RRDs of all samples for SRS-MDEQ-A3 are consistently smaller than 0.2, contrasting with MDEQ-A3, which features numerous samples with large LADs. Notably, with an increasing number of steps, both MDEQ and SRS-MDEQ exhibit a trend toward increased concentration, with SRS-MDEQ consistently outperforming MDEQ. Besides, we also exhibit the mean RRD values over the whole dataset in Table 22, consistently showing the better performance of our SRS-MDEQ.

## I    Correlation-Eliminated Certification

### I.1    Detailed Illustration

In this appendix, we delve deeper into the nuances of our approach to correlation-eliminated certification. As shown in Figure 5, the correlation-eliminated certification is based on dropping unreliable predictions. To be specific, our method compares the predictions of SRS-DEQ with the ones with standard DEQ and then drops the inconsistent predictions in the most probable class $\hat{c}_A$. This process intuitively reassigns all incorrectly classified predictions from class $\hat{c}_A$ to class $\hat{c}_B$, effectively aligning them with the predictions made by the standard DEQ. Finally, we estimate $\overline{p_m}$ to get a conservative estimation of these converted samples.

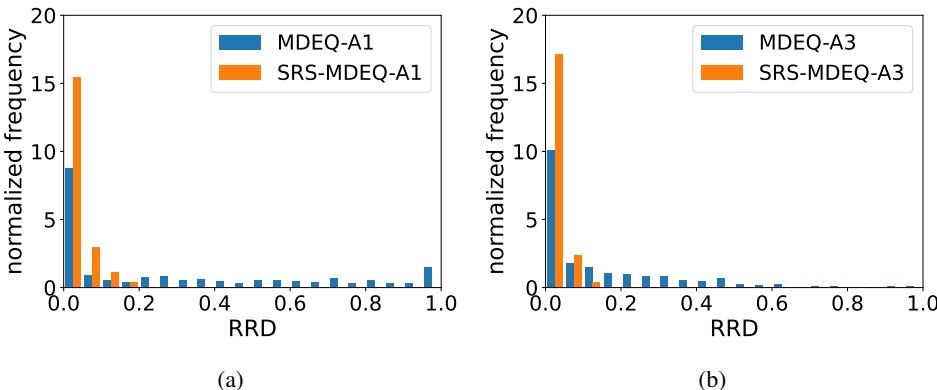

Figure 4: RRD histogram with MDEQ-SMALL models. There are 10 bins in each histogram.

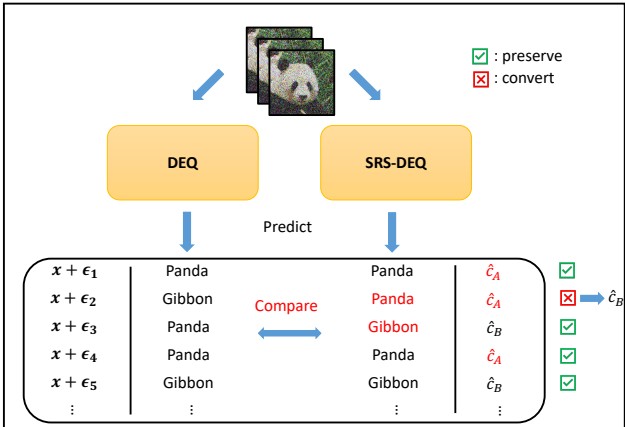

Figure 5: The illustration of our correlation-eliminated certification. If we input the noisy panda images into the standard DEQ and our SRS-DEQ, there will be some misalignment due to the correlation introduced by SRS. Our method conservatively converts these predictions back to the correct ones. For instance, the predictions of $\mathbf{x} + \epsilon_2$ are different with RS and SRS. Therefore, the prediction of $\mathbf{x} + \epsilon_2$ will not be counted as the most probable class $\hat{c}_A$. Finally, we use these converted predictions to calculate the certified radius to recover the standard DEQ's predictions. In the implementation, we try to estimate the number of these converted predictions instead of using the standard DEQ to get the inferences.

## I.2 Cut-off Radius

Given the noise variance $\sigma$, a sampling number $N$, and failure tolerance $\alpha$, the cut-off radius means the maximum radius that can be certified, i.e., the radius when all samples are classified correctly. With SRS, since we have an upper bound on $p_m$, the maximum empirical confidence $p_A$ could be lower. Here we provide analysis for the cut-off radius comparison. To be specific, we present the comparison of the radius with different correct ratios (the percentage of class A) when there are no wrong predictions from our method ($p_m \approx 0$). More formally,

$$R^{srs}_{ratio} = \text{LowerConfBound}((1 - \overline{p}_m)N \times \text{ratio}, N, \tilde{\alpha}) \tag{24}$$

$$\overline{p}_m = \text{LowerConfBound}(B, B, \tilde{\alpha}) \tag{25}$$

We provide the numerical cut-off radius with the hyperparameters used in our paper: $N = 10,000$, $B = 1,000$, $\alpha = 0.001$. The results are shown in Table 23. As a special case (when the ratio is 1), the cut-off radius of our method is 1.594, and the cut-off radius of the original method is 1.860. For the samples with smaller ratios, the gap between the standard method and our method will further decrease because of the marginal effect (Cohen et al., 2019).

| $ratio$ | 0.75 | 0.80 | 0.85 | 0.90 | 0.95 | 0.99 | 1.00 |
|---|---|---|---|---|---|---|---|
| $R_{ratio}^{base}$ | 0.332 | 0.415 | 0.512 | 0.634 | 0.814 | 1.148 | 1.860 |
| $R_{ratio}^{srs}$ | 0.331 | 0.414 | 0.511 | 0.633 | 0.812 | 1.139 | 1.594 |

Table 23: Comparison of $R_{ratio}^{base}$ and $R_{ratio}^{srs}$ for different $ratio$ values.

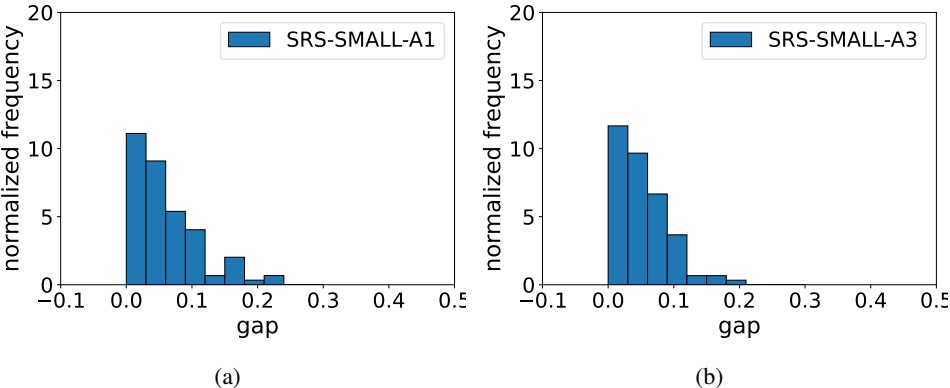

(a)  (b)

Figure 6: Gap histogram with MDEQ-SMALL models. There are 10 bins in each histogram.

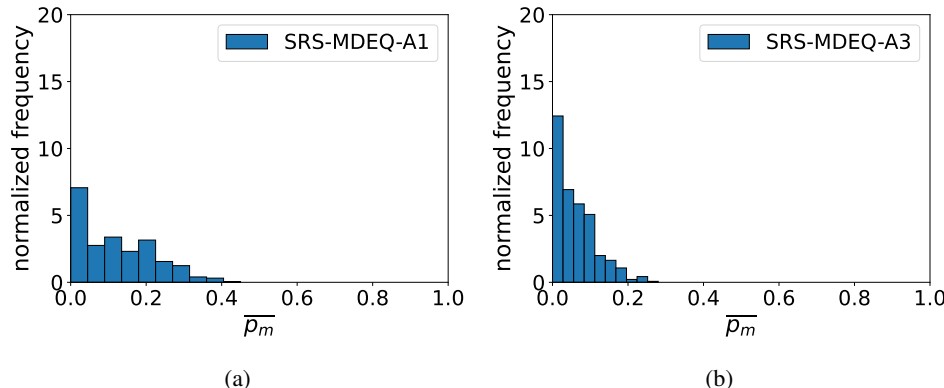

(a)  (b)

Figure 7: $\overline{p_m}$ histogram with MDEQ-SMALL models. There are 10 bins in each histogram.

## I.3 More Results

In this appendix, we extend our analysis to include additional results for the correlation-eliminated certification, focusing particularly on the distribution of the gap and $\overline{p_m}$ for MDEQ-SMALL models. These results are illustrated in Figure 3 and Figure 7. Regarding the gap, we observe a trend consistent with that for MDEQ-LARGE models: a predominant skew towards 0 while maintaining positive values, which underscores the efficacy of our estimation approach. As for $\overline{p_m}$, its distribution appears more uniform in MDEQ-SMALL with one step, compared to MDEQ-LARGE. This aligns with the observed phenomenon where the certified accuracy is somewhat lower than that achieved through standard randomized smoothing for DEQs with a single step.

## J The Number of Samplings

We investigate the effect of the number of samplings in our randomized smoothing approach by conducting experiments with $\sigma = 0.5$ depicted in Figures 8 and 9. Notably, these results align well with those reported in (Cohen et al., 2019). Across all results, a consistent trend emerges, revealing that there are no substantial differences observed between $N = 10,000$ and $N = 100,000$ across most radii. This insight underscores the robustness and stability of the results, emphasizing that the choice of the number of samplings within this range does not significantly impact the outcomes.

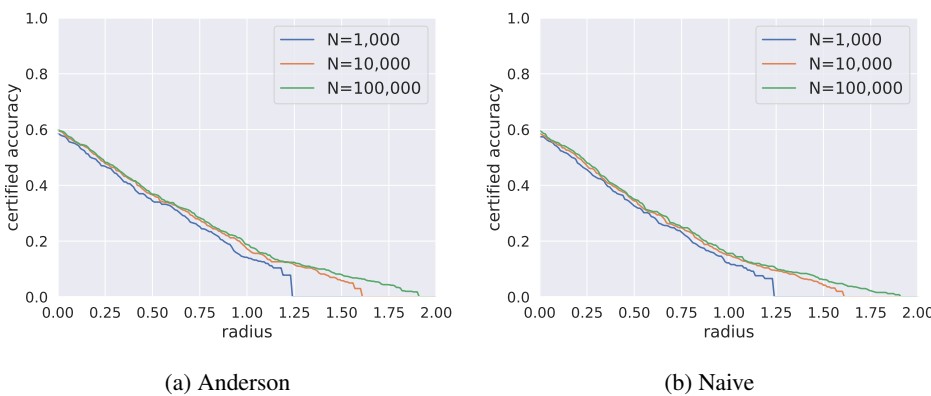

(a) Anderson            (b) Naive

Figure 8: Different number of samplings for MDEQ-SMALL with the 3-step solvers.

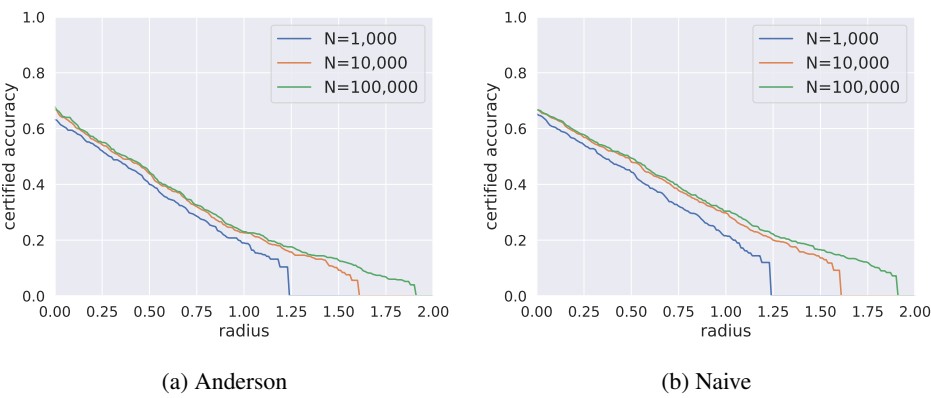

(a) Anderson            (b) Naive

Figure 9: Different number of samplings for MDEQ-LARGE with the 3-step solvers.

## K   Warm-up Strategy

In our implementation, we employ a warm-up strategy to enhance the initial performance of serialized randomized smoothing during certification. In this appendix, we delve into the effectiveness of this technique.

**The Number of Steps:** We investigate the influence of the number of warm-up steps on the performance of our method. As shown in Figures 10a and 10b, there is a marginal improvement in the performance of MDEQ-LARGE when the number of warm-up steps is increased to 30. The performance of MDEQ-SMALL remains stable. For the sake of time efficiency, we adopt 10 as the default parameter in our main experiments.

**The Warm-Up Solver:** We explore whether utilizing different solvers during the warm-up phase impacts the performance of our method, denoting the model with the "solver-solver" format. For example, "Anderson-Naive" signifies the warm-up solver as Anderson and the solver for MDEQ as the naive one. Conducting experiments with 10 warm-up steps and 3-step solvers, the results in Figures 11a and 11b indicate that the choice of warm-up solvers does not significantly affect performance when the solvers for MDEQ are the same (as evidenced by the nearly overlapping lines) for both large and small models. In our main experiments, we consistently use pairwise solvers, where the solver for warming up aligns with that used for MDEQ.

**Restart:** Considering the potential accumulation of fixed-point errors due to the distance of samplings, we investigate the necessity of a restart strategy for our method. Specifically, we implement this strategy by warming up every $K$ batches. Employing pairwise solvers with 10 warm-up steps, the results in Figures 12a and 12b exhibit the findings from the warm-up steps. Implementing a restart strategy with varying intervals yields a slight performance increase. For the sake of time efficiency, we default to 10 as the parameter in our main experiments.

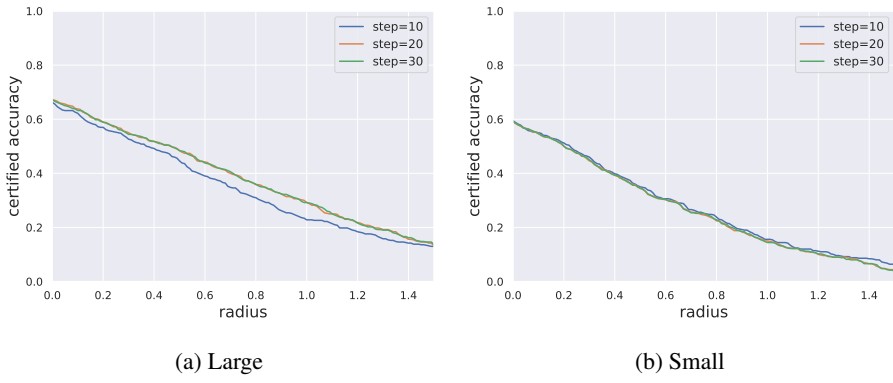

(a) Large           (b) Small

Figure 10: Different warm-up steps for SRS-MDEQ-3A.

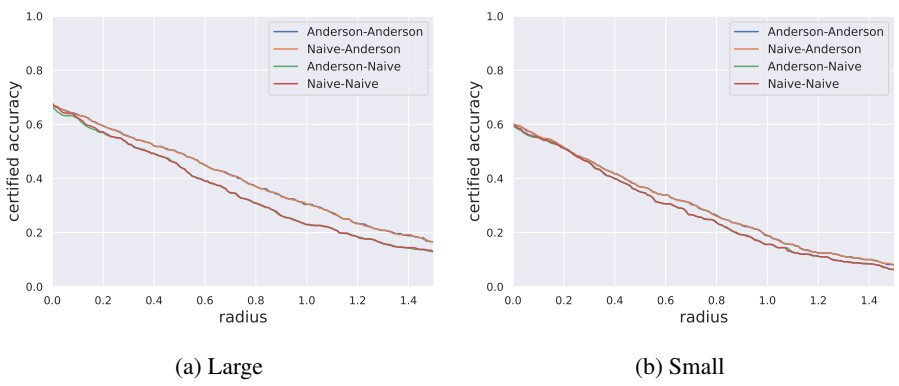

(a) Large           (b) Small

Figure 11: Different warm-up solvers for SRS-MDEQ-3A.

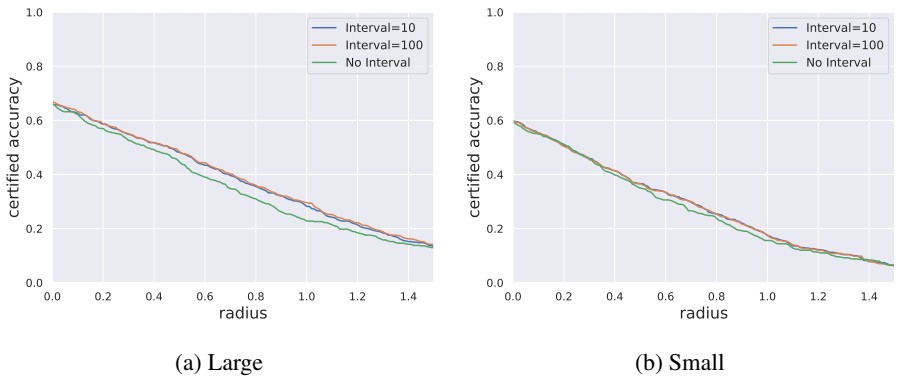

(a) Large           (b) Small

Figure 12: Different warm-up restart intervals for SRS-MDEQ-3A.

**Start Points:** There are two choices for the start points of the warm-up strategy: (1) start from the clean data point $\mathbf{x}$ for all the noisy data $\mathbf{x} + \epsilon$; (2) start from previous noisy data in each batch. We choose to start from the previous noisy data because it performs better than using the fixed-point solution of the clean data. This is because the previous noisy data provides smaller $p_m$ in the correlation-eliminated certification. The certified accuracy is shown in Table 24. The differences between the two start initialization come from the correlated-elimination certification. The final radius will depend on the estimated $p_m$ (smaller $p_m$ is better). With the previous fixed point, our certification process accumulates randomness to avoid the bad guess at the beginning. The distribution of $p_m$ will be more concentrated to 0 with our method that starts from the previous solutions. It means we need to drop more predictions if we start from the clean data. As a result, the certified accuracy of starting from the previous fixed points is better.

| Model \ Radius | 0.0 | 0.25 | 0.5 | 0.75 | 1.0 | 1.25 | 1.5 |
|---|---|---|---|---|---|---|---|
| SRS-MDEQ-1A-clean | 56% | 48% | 40% | 29% | 20% | 16% | 12% |
| SRS-MDEQ-3A-clean | 64% | 52% | 45% | 33% | 23% | 15% | 11% |
| SRS-MDEQ-1A | 63% | 53% | 45% | 32% | 22% | 16% | 12% |
| SRS-MDEQ-3A | 66% | 54% | 45% | 33% | 23% | 16% | 11% |

Table 24: Certified accuracy for the MDEQ-LARGE architecture with $\sigma = 0.5$ on CIFAR-10. The first two rows represent the results starting from clean data, while the latter two rows represent the results starting from the previous fixed-point solutions.

## L   Empirical Robustness

In this appendix, we present the performance of SRS-MDEQ under adversarial attacks to show our model is robust empirically. To demonstrate the efficacy of our approach, we assess predictions using the strongest adversarial attack PGD-$\ell_2$ and its variant for randomized smoothing, Smooth-PGD (Salman et al., 2019).

| Attack | $r = 0.0$ | $r = 0.25$ | $r = 0.5$ | $r = 0.75$ | $r = 1.0$ | $r = 1.25$ | $r = 1.5$ |
|---|---|---|---|---|---|---|---|
| PGD | 72% | 70% | 65% | 63% | 60% | 60% | 58% |
| m=1 | 72% | 67% | 63% | 62% | 61% | 61% | 61% |
| m=4 | 72% | 66% | 62% | 60% | 57% | 57% | 58% |
| m=8 | 72% | 66% | 61% | 58% | 56% | 55% | 54% |
| m=16 | 72% | 65% | 60% | 55% | 53% | 51% | 49% |
| Certified | 66% | 54% | 45% | 33% | 23% | 16% | 11% |

Table 25: The empirical performance of the randomized smoothing on LARGE-SRS-MDEQ-3A.

| Attack | $r = 0.0$ | $r = 0.25$ | $r = 0.5$ | $r = 0.75$ | $r = 1.0$ | $r = 1.25$ | $r = 1.5$ |
|---|---|---|---|---|---|---|---|
| PGD | 66% | 63% | 61% | 57% | 53% | 49% | 45% |
| m=1 | 66% | 61% | 53% | 46% | 40% | 36% | 30% |
| m=4 | 66% | 61% | 53% | 46% | 40% | 36% | 32% |
| m=8 | 66% | 61% | 53% | 46% | 39% | 36% | 30% |
| m=16 | 66% | 61% | 53% | 44% | 39% | 35% | 29% |
| Certified | 60% | 50% | 38% | 29% | 21% | 12% | 8% |

Table 26: The empirical performance of the randomized smoothing on SMALL-SRS-MDEQ-3A.

Projected Gradient Descent (PGD) leverages the principles of gradient descent to iteratively update input data. It begins with an initial input, calculates the gradient of the model's loss concerning the input, and adjusts the input in the direction that maximizes the increase in the loss. This adjustment is constrained by a small perturbation limit to ensure that the changes remain within acceptable bounds. In the context of randomized smoothing, PGD is employed to directly target the base classifier. We utilize a fixed step size of 0.1 for each iteration, and the total number of iterations is set at 20.

Given that PGD does not directly target the smoothed classifier, we also employ Smooth-PGD to attack our model, following the methodology outlined in (Salman et al., 2019). The indirect attack proves ineffective due to the obfuscated gradient phenomenon (Athalye et al., 2018). Smooth-PGD initially utilizes a soft version to approximate the gradient of the smoothed classifier, mitigating the non-differentiable nature of the classifier. It then employs the Monte Carlo method to estimate the value of the gradient. Finally, standard PGD is employed to generate adversarial examples using the estimated gradient. Smooth-PGD demonstrates increased effectiveness compared to PGD when given sufficient samplings in Monte Carlo. Throughout our experiments, we maintain consistency in hyperparameter use between Smooth-PGD and standard PGD.

Following work (Salman et al., 2019), we select the number of samplings $m$ in Smooth-PGD from {1, 4, 8, 16}. We compare the empirical results with the certified accuracy reported in our main paper to

| Attack | $r = 0.25$ | $r = 0.5$ | $r = 0.75$ | $r = 1.0$ | $r = 1.25$ | $r = 1.5$ |
|--------|-----------|----------|-----------|----------|-----------|----------|
| PGD | 5% / 0% | 14% / 0% | 15% / 0% | 17% / 0% | 15% / 0% | 17% / 0% |
| m=1 | 10% / 0% | 16% / 0% | 16% / 0% | 15% / 0% | 14% / 0% | 13% / 0% |
| m=4 | 12% / 0% | 20% / 0% | 20% / 0% | 20% / 0% | 18% / 0% | 16% / 0% |
| m=8 | 14% / 0% | 21% / 0% | 23% / 0% | 23% / 0% | 21% / 0% | 21% / 0% |
| m=16 | 16% / 0% | 23% / 0% | 27% / 0% | 26% / 0% | 27% / 0% | 26% / 0% |

Table 27: The point-wise successful attack rate of on LARGE-SRS-MDEQ-3A. The first number is the rate of successfully attacking the uncertified points. The second number is the rate of successfully attacking the certified points.

| Attack | $r = 0.25$ | $r = 0.5$ | $r = 0.75$ | $r = 1.0$ | $r = 1.25$ | $r = 1.5$ |
|--------|-----------|----------|-----------|----------|-----------|----------|
| PGD | 6% / 0% | 8% / 0% | 13% / 0% | 16% / 0% | 20% / 0% | 23% / 0% |
| m=1 | 10% / 0% | 22% / 0% | 28% / 0% | 32% / 0% | 35% / 0% | 39% / 0% |
| m=4 | 10% / 0% | 22% / 0% | 28% / 0% | 32% / 0% | 34% / 0% | 37% / 0% |
| m=8 | 9% / 0% | 21% / 0% | 28% / 0% | 33% / 0% | 35% / 0% | 39% / 0% |
| m=16 | 10% / 0% | 22% / 0% | 30% / 0% | 33% / 0% | 36% / 0% | 40% / 0% |

Table 28: The point-wise successful attack rate of on SMALL-SRS-MDEQ-3A. The first number is the rate of successfully attacking the uncertified points. The second number is the rate of successfully attacking the certified points.

show the correctness of the randomized smoothing. The results are shown in Tables 25 and 26, where $r$ represents the attack budget. $r = 0$ means the predicted accuracy on clean data. It is observed that Smooth-PGD exhibits superior strength compared to the standard PGD. The most important finding is that the certified accuracy is lower than the accuracy under all the adversarial attacks, meaning all the attacks can not break the certified robustness in randomized smoothing. In essence, these results underscore the robustness of our method, showcasing reliable certified accuracy empirically.

Based on the preceding analysis, certified accuracy serves as a global metric for evaluating model robustness. However, to substantiate the efficacy of certification, the model must ensure that each certified point remains invulnerable within its corresponding certified radius. This appendix conducts an instance-level analysis to demonstrate this aspect. Specifically, we quantify the percentage of points successfully attacked within the certified and uncertified subsets, respectively. The outcomes are detailed in Tables 27 and 28. The first number is the rate of successfully attacking the uncertified points, while the second number is the rate of successfully attacking the certified points. As $m$ increases, the first number gets larger, meaning the attack is getting stronger. In this case, the second number consistently keeps as 0, meaning the certified points can not be attacked at all. Despite the minimal failure probability in randomized smoothing certification, our point-wise empirical investigation underscores the robustness and reliability of our method.

