# OpenReview forum: "Certified Robustness for Deep Equilibrium Models via Serialized Random Smoothing"
_NeurIPS.cc/2024/Conference — NeurIPS 2024 poster_

### Official Review · Reviewer_qqXj · 2024-06-12

**Soundness:** 4
**Presentation:** 3
**Contribution:** 3
**Rating:** 7
**Confidence:** 5

**Summary:**

This work first studies Randomized Smoothing in Deep Equilibrium Models. To combat the prohibitive computational cost, this work designs a procedure called SRS to speed up certification, mainly relying on the fast convergence of multiple predictions with DEM. The certification theorem is revised accordingly to adapt to the new procedure. Experimental results prove that the new algorithm achieves stronger or comparable performance over the standard RS, with a major speedup.

**Strengths:**

This paper first studies RS in DEM, and thus presents a solid step. I appreciate the authors' considering the speedup of the standard approach and the performance of their approach. The theoretical analysis seems correct, despite minor issues.

**Weaknesses:**

I don't find major drawbacks in the approach and the evaluations. Below are some minor concerns:

1. In the evaluation, the authors only show the discretized certified radius curve, while the average certified radius (ACR) is the standard metric for evaluating RS. I suggest the authors to report ACR as well.
2. The authors indicate that the models are trained via Gaussian, while there are many recent advances in training RS models, e.g., SmoothAdv [1] and CATRS [2]. I advise the authors to include the performance of SRS under at least one of these more advanced training tricks, if applicable, to show the universality of their approach w.r.t. the model.
3. Line 190 says that the bounding subset is selected during the SRS sampling, this means the hypothesis test here is not independent of the SRS; thus in the proof, the multiplication of events could fail. The authors should rethink and potentially fix their proof accordingly.
4. SRS basically reduces the complexity of fixed point iteration, by making the prediction serial. However, this is not ideal for parallelization. In Fig 1.b, SRS sequentially uses the result from the last prediction as the initialization; while these factual more precise fixed points could be beneficial, to facilitate parallelization, I suspect that even initializing all others with the result of the first prediction is sufficient, as they follow the same distribution. Could the authors elaborate more on this?

[1] Salman et al., Provably Robust Deep Learning via Adversarially Trained Smoothed Classifiers.
[2] Jeong et al., Confidence-aware Training of Smoothed Classifiers for Certified Robustness.

**Questions:**

See weakness. Below are some additional questions:

Minor:
Line 98 & 158, Cohen et al did not discuss fixed point solvers and DEQs. Is that a wrong citation?
Line 454 & 455, `for $\|\delta\|<R$` is duplicated. Further, I advise the authors to explain why they specifically chose $N^E_A$ in their algorithm: the proof does not seem to motivate this choice. Regarding $p_m$, should it be the probability that the prediction is correct under SRS but not normal RS?

**Limitations:**

The authors do not seem to include a limitation section. They point to the conclusion section in the checklist, which I don't find valid.

---

> ### Author Rebuttal · Authors · 2024-08-05
>
> Dear reviewer, thank you so much for your valuable comments and recognition of the novelty, effectiveness, and presentation of our work. We are happy to address your concerns and questions with the following results and illustrations.
>
> **Weakness 1**: In the evaluation, the authors only show the discretized certified radius curve...
>
> **Answer**: Thank you for your suggestion. We produce the results with the ACR metric for CIFAR-10 following the definition in [1]. Here we report them in Table 1. We will add them to the main paper in a revised version.
>
> | Model                | ACR  |\| Model                | ACR  |
> |----------------------|------|----------------------|------|
> | MDEQ-LARGE-1A        | 0.27 |\| MDEQ-SMALL-1A        | 0.23 |
> | MDEQ-LARGE-30A       | 0.62 |\| MDEQ-SMALL-30A       | 0.59 |
> | SRS-MDEQ-LARGE-1A    | 0.59 | \|SRS-MDEQ-SMALL-1A    | 0.56 |
> | SRS-MDEQ-LARGE-3A    | 0.62 |\| SRS-MDEQ-SMALL-3A    | 0.59 |
>
> Table 1: Average certified radius for the MDEQ architecture with $\sigma=0.5$ on CIFAR-10.
>
> [1] Chen, Ruoxin, et al. "Input-specific robustness certification for randomized smoothing." AAAI 2022.
>
> **Weakness 2**: The authors indicate that the models are trained via Gaussian, while there are many recent advances...
>
> **Answer**: Thank you for your suggestion. Here we produce the results with SmoothAdv to show the effectiveness of our method, and we will add them to the main paper in a revised version. For SmoothAdv, we choose PGD [1] as the adversarial attack and the number of adversarial examples in the training is set as 4. The results are shown in Table 2. With SmoothAdv, the certified accuracy will increase for both the standard randomized smoothing and our SRS.
>
> | Model \ Radius          | 0.0 | 0.25 | 0.5 | 0.75 | 1.0 | 1.25 | 1.5 |
> |-------------------------|-----|------|-----|------|-----|------|-----|
> | MDEQ-30A (adv)          | **62%** | **54%** | 43% | **37%** | **30%** | **23%** | 14% |
> | MDEQ-30A (standard)     | 62% | 50%  | 38% | 30%  | 22% | 13%  | 9%  |
> | SRS-MDEQ-1A (adv)       | 60% | 43%  | 35% | 27%  | 18% | 14%  | 9%  |
> | SRS-MDEQ-3A (adv)       | 60% | 52%  | **43%** | 36%  | 29%  | 22%  | **14%** |
>
> Table 2: Certified accuracy for the MDEQ-SMALL architecture with $\sigma=0.5$ on CIFAR-10 using SmoothAdv.
>
> [1] Kurakin, Alexey, Ian Goodfellow, and Samy Bengio. "Adversarial machine learning at scale." arXiv:1611.01236 (2016).
>
> **Weakness 3**: Line 190 says that the bounding subset is selected during the SRS sampling, this means ...
>
> **Answer**: Thanks for pointing out our problem. Because the two hypothesis tests are both based on noisy samples, they could be dependent. In this case, we slightly revise our proof as follows: denote the event that the radius of SRS is smaller than the radius of RS as $A$ and the event that the radius of RS can certify the data points $B$. We can conclude that $\mathbb{P}(\bar{A})=\mathbb{P}(\bar{B})=\tilde{\alpha}$ following the hypothesis tests. The final probability of successfully certifying the data point is:
> $$
>     \mathbb{P}(A\cup B) = \mathbb{P}(B)-\mathbb{P}(\bar{A}\cup B) = 1-\mathbb{P}(\bar{B})-\mathbb{P}(\bar{A}\cup B) \geq 1-\mathbb{P}(\bar{B})-\mathbb{P}(\bar{A}) = 1-2\tilde{\alpha}
> $$
> where $\mathbb{P}(\bar{A})$ is the probability that $A$ does not happen. By setting $\tilde{\alpha}=\alpha/2$, we complete the proof. By carefully reevaluating the differences between the previous $\tilde{\alpha}=1-\sqrt{1-\alpha}$ and the current one in our proof, the reported accuracy does not change because $\alpha$ is very small (e.g., $\alpha=0.01$). We will revise our proof and the corresponding algorithm in a new version of our paper.
>
> **Weakness 4**: SRS basically reduces the complexity of fixed point iteration, by making the prediction serial...
>
> **Answer**: Because of the page limitation, the corresponding analysis is in Appendix K (start points). For convenience, we briefly restate our conclusion of the analysis. The results with the last fixed points are better than those with the first prediction when the number of the fixed-point iterations is small as shown in Table 2. With the previous fixed point, our certification process accumulates randomness to avoid the bad guess at the beginning so we get better results. We will discuss the question more in the revised version.
>
> | Model \ Radius      | 0.0  | 0.25 | 0.5  | 0.75 | 1.0  | 1.25 | 1.5  |
> |---------------------|------|------|------|------|------|------|------|
> | SRS-MDEQ-1A-clean   | 56%  | 48%  | 40%  | 29%  | 20%  | 16%  | 12%  |
> | SRS-MDEQ-3A-clean   | 64%  | 52%  | 45%  | 33%  | 23%  | 15%  | 11%  |
> | SRS-MDEQ-1A         | 63%  | 53%  | 45%  | 32%  | 22%  | 16%  | 12%  |
> | SRS-MDEQ-3A         | 66%  | 54%  | 45%  | 33%  | 23%  | 16%  | 11%  |
>
> Table 3: Certified accuracy for the MDEQ-LARGE architecture with $\sigma=0.5$ on CIFAR-10. The first two rows represent the results starting from clean data.
>
> **Minor Question:** Line 98 \& 158, Cohen et al did not discuss fixed point solvers and DEQs. Is that a wrong citation?...
>
> **Answer**: Thanks for pointing out the wrong citation, we are supposed to cite the paper about the solver [1]. Besides, we will remove the duplicated words in the text.
>
> As you understand, $p_m$ is the probability that the prediction is correct under SRS but not normal RS. Therefore $N_E^A$ is the number of effective predictions in SRS, which is a conservative estimation of randomized smoothing. We will correct all of those mistakes and clarify the notations in a revised version of our paper.
>
> [1] Bai, Shaojie, Vladlen Koltun, and J. Zico Kolter. "Neural deep equilibrium solvers." ICLR 2021.
>
> **Updating Limitations**: Thanks for pointing out the lack of limitations. We will add the following text in a revised version: ''Though our paper speeds up the certification of DEQs with randomized smoothing, it cannot be directly applied to other architecture. We regard the speedup for the general method as our future research.''

---

> > ### Comment · Reviewer_qqXj · 2024-08-08
> >
> > Thanks for the detailed rebuttal from authors. This clears my concerns. I will raise my score to accept.

---

> > > ### Author Response · Authors · 2024-08-12
> > > **Thank you**
> > >
> > > Thank you very much for reconsidering our submission. We will promptly incorporate the related works you mentioned.
> > >
> > > If you have any further concerns, please don't hesitate to let us know.

---

### Official Review · Reviewer_eMdN · 2024-07-12

**Soundness:** 3
**Presentation:** 3
**Contribution:** 3
**Rating:** 7
**Confidence:** 4

**Summary:**

Due to the inability of existing deterministic methods for providing certified robustness to Deep Equilibrium Models (DEQs) to be applied to large-scale datasets and network structures, this paper provides scalable certified robustness for DEQs through the probabilistic method of random smoothing. To avoid the high computational costs associated with directly using existing random smoothing methods, this paper implements Serialized Randomized Smoothing using historical information from previous inputs to reduce computational overhead. The paper also provides theoretical correctness proofs for this method. Experiments show that significant computational performance improvements can be achieved without sacrificing certified accuracy.

**Strengths:**

Originality. Through an analysis of the computational efficiency of directly applying random smoothing methods to obtain certified robustness for DEQs, it was concluded that the Monte Carlo estimation in the random smoothing method and the fixed-point solver in DEQs are the computational efficiency bottlenecks. This paper proposes a serialized smoothing method, which improves the computational efficiency of the certified robustness method while maintaining certified accuracy. Compared to directly applying random smoothing for certified robustness, this represents a significant improvement in computational performance.

Quality. The paper concisely utilizes the historical feature representation information from other noisy samples, eliminating the substantial redundant computations caused by Monte Carlo estimation in random smoothing. This enhances the practicality of the proposed certified robustness method. Furthermore, by introducing a new certified radius estimation method, they ensure the correctness of this concise algorithm.

Significance. This paper effectively addresses the practicality issues caused by the computational overhead when applying the seemingly universal random smoothing method to specific deep learning models. It provides a concise processing solution tailored to the computational characteristics of specific network models. The authors also conduct a theoretical analysis of the method's correctness resulting from its application. This work offers an inspiring research approach for the research community to improve the computational efficiency of random smoothing in future studies.

**Weaknesses:**

In the process of correlation-eliminated certification, this paper requires the use of standard DEQs to drop unreliable predictions, which necessitates additional memory overhead for standard DEQs during actual deployment.

**Questions:**

In this paper, the authors have been exploring the introduction of a new serialized random smoothing certification method for robustness to avoid the expensive computational costs of previous methods while ensuring that certified accuracy is not affected. I would like to know if the serialized smoothing certification method for robustness has any impact on the clear accuracy metric for DEQs compared to the previous traditional random smoothing certification methods, and if so, how significant is this impact?

**Limitations:**

yes

---

> ### Author Rebuttal · Authors · 2024-08-05
>
> Dear reviewer, thank you so much for your valuable comments and recognition of the novelty, effectiveness, and presentation of our work. We are happy to address your concerns and questions.
>
> **Weakness 1**: In the process of correlation-eliminated certification, this paper requires the use of standard DEQs to drop unreliable predictions, which necessitates additional memory overhead for standard DEQs during actual deployment.
>
> **Answer**: We think there are some slight misunderstandings of the DEQs. The memory cost of the standard DEQs is independent of the number of the fixed-point iterations. Our experiments show that both of them use about 1.5 GB of memory with a batch size of 400. The standard DEQs have the same memory overhead as our method because we only change the number of fixed-point iterations but keep the same solvers. Therefore, we only need more time to run the standard randomized smoothing. Thanks for your question and we will clarify it clearly in the revision of our paper.
>
> **Question 1**: In this paper, the authors have been exploring the introduction of a new serialized random smoothing certification method for robustness to avoid the expensive computational costs of previous methods while ensuring that certified accuracy is not affected. I would like to know if the serialized smoothing certification method for robustness has any impact on the clear accuracy metric for DEQs compared to the previous traditional random smoothing certification methods, and if so, how significant is this impact?
>
> **Answer**: We assume you are referring to the ''clean'' accuracy of our proposed method instead of the ''clear'' accuracy. In Table 1-3 of the main paper, we show the certified accuracy under different radii. The clean accuracy is a special case where the radius is 0. In this case, we do not require the smoothed classifier to be robust but to have the correct predictions. For convenience, we copy part of the results to show it. Our method will sacrifice little clean accuracy (only 1\%) compared to the standard randomized smoothing as shown in Table 1.
>
> Hope our answer can address your question, and we are glad to reply if you need more description.
>
> | Model       | Clean Accuracy |
> |-------------|----------------|
> | MDEQ-30A    | 67%            |
> | SRS-MDEQ-3A | 66%            |
>
> Table 1: Clean accuracy for the MDEQ-LARGE on CIFAR-10.

---

> > ### Comment · Reviewer_eMdN · 2024-08-13
> >
> > Thank you for the detailed response. This addresses my concerns, so I’ll be raising my score to accept.

---

> > > ### Author Response · Authors · 2024-08-13
> > > **Thank you**
> > >
> > > Thank you very much for your positive feedback and reconsidering our submission.
> > >
> > > If you have any further concerns, please don't hesitate to let us know.

---

### Official Review · Reviewer_366J · 2024-07-12

**Soundness:** 3
**Presentation:** 3
**Contribution:** 3
**Rating:** 7
**Confidence:** 3

**Summary:**

This paper provides a method for randomized smoothing for DEQs. Given the computational challenges associated with DEQs, the authors propose a method that is intended to speed up the process of creating a smooth classifier for a DEQ model based on fixed-point reuse. Given that fixed-point reuse introduces dependency between the predictions for the models f, the authors propose a method called Correlation-Eliminated Certification in which the goal is to remedy the aforementioned issue with fixed-point reuse. This method is centered on approximating the probability with which SRS misclassifies a point as the most probable class, and uses the original DEQ to achieve this. Then the authors provide a high probability certification result in Theorem 3.1.

**Strengths:**

* The paper adapts the certified robustness method of randomized smoothing to deep equilibrium models while keeping in mind the nuances of a DEQ. In doing so, they come up with a novel and unique method for certifying a DEQ. The paper also provides careful analysis of their method, and shows that their method is both efficient and admits high certified accuracy.
* The authors provide a thorough ablations section which answers questions regarding the goodness of their approximation for $p_m$.
* The authors provide a theoretical guarantee for the correctness of their method, and provide extensive experimental results.

**Weaknesses:**

* It would be nice to provide more careful definitions. One definition that should be in the paper is that of $\ell_2$-norm certified radius, as is defined for example in [1]. A pointer to a paper which introduces this is also missing. Generally speaking, although [1] also uses similar notation to this paper, it is a bit confusing to write $c_A$ without writing it as a function of $x$. My understanding is that $c_A$ is the most likely class given some sample $x$, so as a reader this is a bit confusing to not write it as a function of $x$. Another definition which seems to be crucial in your paper is LowerConfBound (e.g. Equation (12)) and hence it would be nice to provide the definition rather than to point the reader to [1]. Furthermore, there are less crucial definitions such as $K$, in Equation (10), which are not defined (but I assume represents the number of samples from SRS, $K < N$). Furthermore, can you please make the notation consistent between Section 3.3 and Appendix A which contains the proof for Theorem 3.1? Specifically there is lower-case $y$ in Appendix A and the subscript notation for the labels is also different between the two sections.
* Using a model with Jacobian regularization seems as though it might be more suitable for a certification method. It would be nice to provide more justification as regards to this choice, and how the method would perform on a MDEQ trained without Jacobian regularization.


[1] Cohen, J., Rosenfeld, E., and Kolter, Z. Certified adversarial robustness via randomized smoothing. In international conference on machine learning, pp. 1310–1320. PMLR, 2019

Some notes:
* Lines 49-54 introduce acronyms that are not defined
* Neyman-Pearson is spelled incorrectly on line 108
* Figure 5 caption: converted -> converts. The sentence starting with “For instance” does not read well. Overall, could you write your caption a bit more clearly?
* Line 574: missing period
* Algorithm 1: line 13 Should say Predict Y_g? It seems as though some indexing is also missing. Also, it is not clear to me what the relationship between the  total number of samples is as the number of samples that is used for $p_m$ in Algorithm 1.

**Questions:**

1. What is the effect of using a model trained with Jacobian regularization on your method? Do you think this is crucial for certifying a DEQ?
2. What is the relationship between the number of samples that are checked with the DEQ in Line 13 of Algorithm 1, and the total number of samples obtained via SRS?

**Limitations:**

Yes.

---

> ### Author Rebuttal · Authors · 2024-08-05
>
> Dear reviewer, thank you so much for your valuable comments and recognition of the novelty, effectiveness, and presentation of our work. We are happy to address your concerns and questions.
>
> **Question 1**: What is the effect of using a model trained with Jacobian regularization on your method? Do you think this is crucial for certifying a DEQ?
>
> **Answer**: Jacobian regularization stabilizes the training of the backbones but it is not crucial for the certification. To answer your question, we conduct the experiments without the Jacobian regularization as shown in Table 1. The results show that using Jacobian regularization can help stabilize the fixed-point solvers but will almost not affect the final performance with enough fixed-point iterations. Generally speaking, the more stable the model is, the more efficient our approach is (i.e., we can use fewer steps of iterations). Our conclusion is consistent with [1] where the regularization does not increase the accuracy but decreases the number of fixed-point iterations.
>
> The experiments show that using Jacobian regularization is not crucial in the certification. However, we recommend to use the regularization in the training for more stable performance. We hope this answer addresses your question, and we are happy to provide further details if needed.
>
> | Model \ Radius         | 0.0 | 0.25 | 0.5 | 0.75 | 1.0 | 1.25 | 1.5 |
> |------------------------|-----|------|-----|------|-----|------|-----|
> | MDEQ-30A (w/o Jacobian) | 63% | 51%  | 38% | 29%  | 19% | 13%  | 7%  |
> | SRS-MDEQ-3A (w/o Jacobian) | 60% | 49%  | 38% | 28%  | 18% | 12%  | 6%  |
> | MDEQ-30A (w Jacobian)  | 62% | 50%  | 38% | 30%  | 22% | 13%  | 9%  |
> | SRS-MDEQ-3A (w Jacobian) | 60% | 50%  | 38% | 29%  | 21% | 12%  | 8%  |
>
> Table 1: Ablation study of Jacobian regularization for the MDEQ-SMALL architecture with $\sigma=0.5$ on CIFAR-10.
>
> [1] Bai, Shaojie, Vladlen Koltun, and Zico Kolter. "Stabilizing Equilibrium Models by Jacobian Regularization." International Conference on Machine Learning. PMLR, 2021.
>
> **Question 2**: What is the relationship between the number of samples that are checked with the DEQ in Line 13 of Algorithm 1, and the total number of samples obtained via SRS?
>
> **Answer**: Thank you for pointing out our unclear expressions and overlooking the equation changes with the notations in Algorithm 1. Generally speaking, Line 13 of Algorithm 1 returns the standard DEQs predictions, which are used to compare with the SRS predictions. The comparison tells us what $p_m$ should be. After dropping the unreliable predictions with $p_m$, we will use the total number of samples $N$ to predict the certified radius.
>
> To be specific, Line 14 should be expanded into two steps. First, we use equation (12) to compute the estimated $\overline{p_m}$, namely:
> $$
> N_1 = \sum\nolimits_{i=1}^{K}\mathbf{1} \\{Y_m = Y_g \text{ and } Y_g = c_A(x) \\} ,
> $$
> $$
> N_2 = \sum\nolimits_{i=1}^{K}\mathbf{1}\\{Y_m = c_A(x)\\},
> $$
> $$
> \overline{p_m} = 1-\text{LowerConfBound}(N_1, N_2, 1-\tilde{\alpha}),
> $$
> Then we use equation (9) to estimate the effective samples that are predicted as class $c_A (x)$ with our estimated $\overline{p_m}$:
> $$
> N_A^E = N_A - \overline{p_m} N_A,
> $$
> Finally, with the total number of samples $N$, we compute the radius with equations (13) and (14):
> $$
> \underline{p_A} = \text{LowerConfBound}(N_A^E, N, 1-\tilde{\alpha})
> $$
> $$
> R = \sigma\Phi^{-1}(\underline{p_A}),
> $$
> We hope this answer addresses your question, and we are happy to provide further details if needed.
>
> **Weakness 1**: Revising the typos and unclear expressions
>
> **Answer**: Thanks for pointing out the inconsistency of the notations and the unclear expressions and we are glad to correct them in a revised version. First, we will add the definition of $\ell_2$ norm and claim $c_A$ as the function of $\mathbf{x}$, namely $c_A (\mathbf{x})$. Secondly, we will correct the typos and the inconsistency notation in the proof. Thirdly, we will add the definition of acronyms for Interval Bound Propagation (IBP) and Lipschitz Bounded Equilibrium Networks (LBEN).
> Besides, we will revise the caption of Fig.5 as: ``For instance, the predictions of $\mathbf{x}+\epsilon_2$ are different with RS and SRS. Therefore, the prediction of  $\mathbf{x}+\epsilon_2$ will not be counted as the most probable class $\hat{c}_A(\mathbf{x})$.''
>
> For the definition of $K$, it is a hyperparameter that we add descriptions but forget to mark it. To be specific, we are supposed to define it from line 189 to line 191: ``During the Monte Carlo sampling of SRS, we randomly select $K$ of samples (a small number compared to $N$) along with their corresponding predictions''.
>
> Finally, we will add the definition of LowerConfBound as follows: LowerConfBound$(k, n, 1-\alpha)$ returns a one-sided $(1-\alpha)$ lower confidence interval for the Binomial parameter $p$ given that $k\sim\text{Binomial}(n, p)$. In other words, it returns some number $\underline{p}$ for which $\underline{p} \leq p$ with probability at least $1-\alpha$ over the sampling of $k\sim\text{Binomial}(n, p)$.
>
> **Weakness 2**: Using a model with Jacobian regularization seems as though it might be more suitable for a certification method. It would be nice to provide more justification as regards to this choice, and how the method would perform on a MDEQ trained without Jacobian regularization.}
>
> **Answer**: To solve your concern, we refer you to the results we provided in Question 1.

---

> > ### Comment · Reviewer_366J · 2024-08-12
> >
> > Thank you for answering my questions and for addressing my concerns. I maintain my recommendation to accept the paper.

---

> > > ### Author Response · Authors · 2024-08-13
> > > **Thank you**
> > >
> > > Thank you very much for your positive feedback. We appreciate your thoughtful review and are glad that our work met your expectations. If you have any additional comments or suggestions, we would be happy to address them.

---

### Official Review · Reviewer_2B8g · 2024-07-14

**Soundness:** 3
**Presentation:** 4
**Contribution:** 4
**Rating:** 7
**Confidence:** 3

**Summary:**

This paper develops the first randomized smoothing certified defense for DEQs, termed as Serialized Random Smoothing (SRS). To address the scalability issue of randomized smoothing, To reduce computational redundancy, SRS leverages historical information and a new certified radius estimation. The proposed method can cover various DEQ structures, significantly expanding the scope of existing work. Extensive experiments and ablation studies on large-scale tasks such as ImageNet have been presented to demonstrate the proposed method. Overall, this paper presents a significant contribution to the field of certified robustness for DEQs. The SRS method offers a promising approach to make certification more practical for these models, especially on larger datasets.

**Strengths:**

1. This paper addresses an important gap in the literature by obtaining non-trivial certified robustness of DEQs across various datasets and network structures.

2. Both the empirical and theoretical developments are solid. The proposed method significantly reduces computation time, making certification feasible for larger models and datasets. The authors have included extensive evaluations on different datasets, model sizes, and hyperparameters.

3. The paper is well written and easy to follow.

**Weaknesses:**

1. The conceptual novelty may not be that significant since randomized smoothing is a well-known technique in the first place.

2. By the end of the paper, it is unclear whether the results in this paper have achieved SOTA certified robustness among all networks. I mean, does the certified robust accuracy of DEQ reach to the level of existing results on feed-forward networks using standard RS?

**Questions:**

1. Can the authors clarify the difficulty and novelty of implementing their method for large-scale datasets like ImageNet?

2. Does the certified robust accuracy of DEQ reach to the level of existing results on feed-forward networks using standard RS?

3. Can the authors further justify the unique novelty and technicality of their contribution?

**Limitations:**

The authors have partially addressed limitations in their paper. However, there seems to lack a substantial discussion on broader societal impacts.

---

> ### Author Rebuttal · Authors · 2024-08-05
>
> Dear reviewer, thank you so much for your valuable comments and recognition of the topic, effectiveness, and presentation of our work. We are happy to address your concerns and questions with the following results and illustrations.
>
> **Question 1**: Can the authors clarify the difficulty and novelty of implementing their method for large-scale datasets like ImageNet?
>
> ***Answer**: For large-scale datasets like ImageNet, the standard randomized smoothing can be too expensive to apply on DEQs because (1) DEQs can be slow on such a high-resolution dataset and (2) need a second-order fixed-point solver to guarantee convergence. With other methods, such as IBP and LBEN, it is hard to compute a non-trivial certified radius because the certification is deterministic [1]. In contrast, our method accelerates the standard randomized smoothing by reusing the historical fixed points and guarantees the theoretical correctness of the certification by the two-stage hypothesis testing.
>
> We hope this answer addresses your question, and we are happy to provide further details if needed.
>
> [1] Li, Linyi, Tao Xie, and Bo Li. "Sok: Certified robustness for deep neural networks." 2023 IEEE symposium on security and privacy (SP). IEEE, 2023.
>
> **Question 2**: Does the certified robust accuracy of DEQ reach to the level of existing results on feed-forward networks using standard RS?
>
> **Answer**: Thanks for your question. To address your concern, we are glad to provide a comparison between the explicit models and DEQs. Despite surpassing the performance of explicit neural networks is not our target, we claim the performance over DEQs can catch up with them, as shown in Table 1. We provide a comparison between DEQs and ResNet-110 under the same training and evaluation setting, and the results are consistent with those reported in [1]. We will add the results to the main paper in a revised version.
>
> | Model\Radius | 0.0 | 0.25 | 0.5 | 0.75 | 1.0 | 1.25 | 1.5 |
> |--------------|-----|------|-----|------|-----|------|-----|
> | ResNet-110   | 65% | 54%  | 41% | 32%  | 23% | 15%  | 9%  |
> | MDEQ-30A     | **67%** | **55%** | 45% | 33% | 23% | 16% | **12%** |
> | SRS-MDEQ-3A  | 66% | 54%  | **45%** | **33%** | **23%** | **16%** | 11% |
>
> Table 1: Certified accuracy for the MDEQ-LARGE architecture with $\sigma=0.5$ on CIFAR-10. The best certified accuracy for each radius is in bold.
>
> [1] Cohen, Jeremy, Elan Rosenfeld, and Zico Kolter. "Certified adversarial robustness via randomized smoothing." international conference on machine learning. PMLR, 2019.
>
> **Question 3**: Can the authors further justify the unique novelty and technicality of their contribution?
>
> **Answer**: The major contribution of this paper is to explore randomized smoothing certification for implicit models for the first time. Specifically, we discover that existing randomized smoothing techniques are not suitable for certifying implicit models such as DEQs because of the dominating computation cost. Therefore, we propose serialized randomized smoothing that leverages the historical fixed points to effectively reduce the computation redundancy and accelerate randomized smoothing significantly. However, the serialized operation brings correlations between the predictions, breaking the correctness of the existing theorem of randomized smoothing. To solve the challenge, we then propose a two-stage certification technique (hypothesis testing) to provide correct certification. The new theorem and empirical studies verify that our algorithm works as expected.
>
> **Weakness 1**: The conceptual novelty may not be that significant since randomized smoothing is a well-known technique in the first place.
>
> **Answer**: Though randomized smoothing is a well-known technique, it cannot be easily applied to DEQs for computation reasons as we illustrate in Question 1. Therefore, we propose our novel serialized randomized smoothing which is much more efficient and with a new theorem as illustrated in Question 3. In this way, our method demonstrates novelty. We hope this answer addresses your question, and we are happy to provide further details if needed.
>
> **Weakness 2**: By the end of the paper, it is unclear whether the results in this paper have achieved SOTA certified robustness among all networks. I mean, does the certified robust accuracy of DEQ reach to the level of existing results on feed-forward networks using standard RS?
>
> **Answer**: To answer your concern, we refer to the comparison provided in Question 2.
>
> **Updating Limitations**: Thanks for pointing out the lack of a substantial discussion on societal impacts. We will add the following discussion in the revised version: ''Our work significantly improves the security of artificial intelligence, especially applicable in sensitive domains. Our proposed SRS can provide significant acceleration of defending the attacks for AI models, enhancing the appliance of the models and maintaining the integrity of AI-driven decisions.''

---

> > ### Comment · Reviewer_2B8g · 2024-08-10
> > **Thanks for the response**
> >
> > Thanks for addressing my comments. I have raised my score to 7.

---

> > > ### Author Response · Authors · 2024-08-12
> > > **Thank you**
> > >
> > > Thank you very much for reconsidering our submission.
> > >
> > > If you have any further concerns, please don't hesitate to let us know.

---

### Official Review · Reviewer_3u5s · 2024-07-16

**Soundness:** 3
**Presentation:** 4
**Contribution:** 3
**Rating:** 7
**Confidence:** 4

**Summary:**

This paper introduces a novel method based on random smoothing to improve the certified robustness of Deep Equilibrium Models (DEQs), a promising type of implicit neural network.

Directly applying random smoothing to DEQs incurs high computational costs. To overcome this issue, the authors leverage the properties of DEQs and design a serialized random smoothing (SRS) strategy. This method uses the output from one noisy input as the starting point for the next, significantly accelerating convergence.

Subsequently, to eliminate the dependence between noisy samples created by SRS, the authors introduce Correlation-Eliminated Certification, a new method to obtain certified radius estimation.

Experimental results on CIFAR-10 and ImageNet demonstrate the efficiency of the proposed model.

**Strengths:**

- The paper is well-written, making it easy to follow the key techniques. The experiments and ablation studies are informative.
- The proposed method presents an inspiring use of the representational properties of DEQs. Specifically, DEQs characterize their output as the fixed point of a function (conditioned on their input). Therefore, when running inference on a group of similar samples, DEQs are essentially solving a group of similar fixed-point equations; their inference can be accelerated by reusing the fixed-point between samples.
- The designed certified radius estimation algorithm, Correlation-Eliminated Certification, is straightforward but effective. With a two-stage hypothesis test, this method eliminates the dependency (a side effect of SMS).

**Weaknesses:**

The proposed method is only applicable to DEQs. Although DEQs are a promising architecture, they have not been as widely applied as explicit neural networks or proven to scale well, limiting the scope of this paper. Moreover, the paper does not include a performance comparison of the proposed against other non-DEQ models.

**Questions:**

- On line 52, "Due to the conservative certification, IBP and LBEN cannot be generalized to large-scale datasets (e.g., ImageNet)." There seems to be a logical gap in this sentence. Could the authors elaborate more on this point? Is there any reference that can back up this claim?
- Is the certified robustness of the proposed model (SRS-DEQ) competitive against the certified robustness of other non-DEQ architectures with random smoothing? For example, the paper "CERTIFIED ROBUSTNESS FOR DEEP EQUILIBRIUM MODELS VIA INTERVAL BOUND PROPAGATION" compares their models with other explicit neural networks. Such comparisons are important for the audience to position the proposed model in a larger context.
- Are there any insights or techniques from this paper that can be applied to improve other performance aspects of DEQs (besides certified robustness) or improve non-DEQ models?

---

> ### Author Rebuttal · Authors · 2024-08-05
>
> Dear reviewer, thank you so much for your valuable comments and recognition of the novelty, effectiveness, and presentation of our work. We are happy to address your concerns and questions with the following results and illustrations.
>
> **Question 1**: On line 52, "Due to the conservative certification, IBP and LBEN cannot be generalized to large-scale datasets (e.g., ImageNet)." There seems to be a logical gap in this sentence. Could the authors elaborate more on this point? Is there any reference that can back up this claim?
>
> **Answer**: Thanks for your question. The sentence here tries to convey that the deterministic methods will generate a trivial certified radius (namely, close to 0) because their estimation is not tight enough in some cases such as deep networks [1]. When it comes to large-scale datasets, we refer to high-resolution and many-class cases, such as ImageNet. It increases the difficulty of the task and usually requires more complex models, deteriorating the trivial certified radius problem. In Section E of [2] (the first paragraph and the practical implications part), the authors make similar claims. We will describe the logic and cite the above reference to support our claim in a revised version.
>
> [1] Zhang, Bohang, et al. "Towards certifying l-infinity robustness using neural networks with l-inf-dist neurons." International Conference on Machine Learning. PMLR, 2021.
>
> [2] Li, Linyi, Tao Xie, and Bo Li. "Sok: Certified robustness for deep neural networks." 2023 IEEE symposium on security and privacy (SP). IEEE, 2023.
>
> **Question 2**: Is the certified robustness of the proposed model (SRS-DEQ) competitive against the certified robustness of other non-DEQ architectures with random smoothing? For example, the paper "CERTIFIED ROBUSTNESS FOR DEEP EQUILIBRIUM MODELS VIA INTERVAL BOUND PROPAGATION" compares their models with other explicit neural networks. Such comparisons are important for the audience to position the proposed model in a larger context.
>
> **Answer**: We appreciate that you recognize the value of DEQs, which present advantages compared to explicit neural networks in some cases, such as memory efficiency in the training and accuracy-speed trade-off during inference [1]. According to your suggestion, we are glad to provide a comparison between the explicit models and DEQs. Despite surpassing the performance of explicit neural networks is not our target, we claim the performance of DEQs can catch up with them, as shown in Table 1. We provide the comparison between DEQs and ResNet-110 under the same training and evaluation setting, and the results are consistent with those reported in [2]. We will add the results to the main paper in a revised version.
>
> | Model\Radius | 0.0 | 0.25 | 0.5 | 0.75 | 1.0 | 1.25 | 1.5 |
> |--------------|-----|------|-----|------|-----|------|-----|
> | ResNet-110   | 65% | 54%  | 41% | 32%  | 23% | 15%  | 9%  |
> | MDEQ-30A     | **67%** | **55%** | 45% | 33% | 23% | 16% | **12%** |
> | SRS-MDEQ-3A  | 66% | 54%  | **45%** | **33%** | **23%** | **16%** | 11% |
>
> Table 1: Certified accuracy for the MDEQ-LARGE architecture with $\sigma=0.5$ on CIFAR-10. The best certified accuracy for each radius is in bold.
>
> [1] Bai, Shaojie, J. Zico Kolter, and Vladlen Koltun. "Deep equilibrium models." Advances in neural information processing systems 32 (2019).
>
> [2] Cohen, Jeremy, Elan Rosenfeld, and Zico Kolter. "Certified adversarial robustness via randomized smoothing." international conference on machine learning. PMLR, 2019.
>
> **Question 3**: Are there any insights or techniques from this paper that can be applied to improve other performance aspects of DEQs (besides certified robustness) or improve non-DEQ models?
>
> **Answer**: Yes. The insights and techniques to exploit computation redundancy can be applied to improve other performance aspects. For instance, the uncertainty estimation with Bayesian inference requires multiple samples of the model parameters, resulting in inference with similar model parameters (instead of similar data input in this paper). When applying the Bayesian neural networks to DEQs, our techniques can improve the efficiency of the uncertainty estimation. However, we must admit that our current fixed-point reusing technique may be not directly applied to non-DEQ models, and new techniques to exploit computation redundancy will need to be developed. This is one of our future work.
>
> **Weakness 1**: The proposed method is only applicable to DEQs. Although DEQs are a promising architecture, they have not been as widely applied as explicit neural networks or proven to scale well, limiting the scope of this paper. Moreover, the paper does not include a performance comparison of the proposed against other non-DEQ models.
>
> **Answer**: We appreciate that you recognize the value of DEQs again. The corresponding results are provided in the reply to Question 2.

---

> > ### Comment · Reviewer_3u5s · 2024-08-09
> >
> > Thank you for addressing all my questions with experiments and clarification. I have raised my score accordingly.
> >
> > I found your response to Question 3 especially inspiring. I think that adding a paragraph in the related work section discussing the “DEQ fixed-point reuse” technique would broaden the appeal of this paper, and effectively position the paper's core technique within its context. For example, besides the Bayesian inference example that you gave, [1] exploited fixed-point reuse in the diffusion process, and [2] applied the technique to optical flow estimation.
> >
> > [1] Bai, Xingjian, and Luke Melas-Kyriazi. "Fixed Point Diffusion Models." CVPR, 2024.
> >
> > [2] Bai, Shaojie, Zhengyang Geng, Yash Savani, and J. Zico Kolter. "Deep Equilibrium Optical Flow Estimation." CVPR, 2024.

---

> > > ### Author Response · Authors · 2024-08-12
> > > **Thank you**
> > >
> > > Thank you very much for reconsidering our submission. We will promptly incorporate the related works you mentioned.
> > >
> > > If you have any further concerns, please don't hesitate to let us know.

---

### Decision · Program_Chairs · 2024-09-25

**Decision:**

Accept (poster)

**Comment:**

This paper advances a new method to achieve certified robustness on deep equilibrium models which reduces what would otherwise be high computational costs. The paper also provides proofs showing correctness of the method. After robust discussion all reviewers agreed the paper was novel, interesting, and overall a good contribution to NeurIPS.

I am recommending this paper be accepted as a poster and not a higher accolade mostly because the scope of the paper is fairly narrow. While it was well-executed, it only focuses on deep equilibrium models which could limit the breadth of its impact.

I recommend the authors to add clear discussions both of the limitations of their work and potential broader impacts for the camera-ready version.